# A novel mode of capping protein-regulation by twinfilin

**Adam B Johnston[1][†], Denise M Hilton[1][†], Patrick McConnell[2], Britney Johnson[3], Meghan T Harris[1], Avital Simone[1], Gaya K Amarasinghe[3], John A Cooper[2], Bruce L Goode[1]***

[1]Department of Biology, Rosenstiel Basic Medical Science Research Center, Brandeis University, Waltham, United States; [2]Department of Biochemistry and Molecular Biophysics, Washington University, St Louis, United states; [3]Department of Pathology and Immunology, Washington University, St Louis, United States

**Abstract** Cellular actin assembly is controlled at the barbed ends of actin filaments, where capping protein (CP) limits polymerization. Twinfilin is a conserved in vivo binding partner of CP, yet the significance of this interaction has remained a mystery. Here, we discover that the C-terminal tail of Twinfilin harbors a CP-interacting (CPI) motif, identifying it as a novel CPI-motif protein. Twinfilin and the CPI-motif protein CARMIL have overlapping binding sites on CP. Further, Twinfilin binds competitively with CARMIL to CP, protecting CP from barbed-end displacement by CARMIL. Twinfilin also accelerates dissociation of the CP inhibitor V-1, restoring CP to an active capping state. Knockdowns of Twinfilin and CP each cause similar defects in cell morphology, and elevated Twinfilin expression rescues defects caused by CARMIL hyperactivity. Together, these observations define Twinfilin as the first 'pro-capping' ligand of CP and lead us to propose important revisions to our understanding of the CP regulatory cycle.

DOI: https://doi.org/10.7554/eLife.41313.001

**\*For correspondence:**
goode@brandeis.edu

[†]These authors contributed equally to this work

**Competing interests:** The authors declare that no competing interests exist.

## Introduction

Assembly of cellular actin structures with distinct architectural and dynamic properties requires the convergence and coordination of numerous actin assembly, stabilization, and disassembly mechanisms. Although our understanding of the functions and mechanisms of individual actin-binding proteins has grown tremendously, there is a need to consider more deeply how seemingly disparate and sometimes competing factors work together in vivo and take on new mechanistic roles within more complex mixtures. One particularly enigmatic example is the interaction of Twinfilin with Capping Protein (CP). These two conserved proteins directly interact with high-affinity, and yet have seemingly opposite effects on the barbed ends of actin filaments.

Twinfilin is one of five proteins in the Actin Depolymerization Factor-Homology (ADF-H) domain family, of which ADF/Cofilin is the founding member (*Poukkula et al., 2011*). Twinfilin is unique among the members of this family in containing two ADF-H domains, which are joined by a small linker region and followed by a short C-terminal tail. Initial biochemical studies categorized Twinfilin as an actin monomer sequestering factor because of its high affinity for ADP-bound G-actin and ability to inhibit subunit addition to either end of the filament (*Goode et al., 1998*; *Vartiainen et al., 2000*; *Wahlström et al., 2001*). However, mouse Twinfilin was later shown to interact directly with the barbed ends of actin filaments (*Helfer et al., 2006*; *Paavilainen et al., 2007*), and more recently yeast Twinfilin was shown to accelerate depolymerization at actin filament ends (*Johnston et al., 2015*). Alone, yeast Twinfilin enhanced barbed end depolymerization by 3-fold through a processive filament end-attachment mechanism. Further, in conjunction with Srv2/CAP (cyclase-associated protein), yeast Twinfilin increased the rate of pointed-end depolymerization by over 15-fold

**eLife digest** Plant and animal cells are supported by skeleton-like structures that can grow and shrink beneath the cell membrane, pushing and pulling on the edges of the cell. This scaffolding network – known as the cytoskeleton – contains long strands, or filaments, made from many identical copies of a protein called actin. The shape of the actin proteins allows them to slot together, end-to-end, and allows the strands to grow and shrink on-demand. When the strands are the correct length, the cell caps the growing ends with a protein known as Capping Protein. This helps to stabilize the cell's skeleton, preventing the strands from getting any longer, or any shorter.

Proteins that interfere with the activity of Capping Protein allow the actin strands to grow or shrink. Some, like a protein called V-1, attach to Capping Protein and get in the way so that it cannot sit on the ends of the actin strands. Others, like CARMIL, bind to Capping Protein and change its shape, making it more likely to fall off the strands. So far, no one had found a partner that helps Capping Protein limit the growth of the actin cytoskeleton.

A protein called Twinfilin often appears alongside Capping Protein, but the two proteins seemed to have no influence on each other, and had what appeared to be different roles. Whilst Capping Protein blocks growth and stabilizes actin strands, Twinfilin speeds up their disassembly at their ends. But Johnston, Hilton et al. now reveal that the two proteins actually work together. Twinfilin helps Capping Protein resist the effects of CARMIL and V-1, and Capping Protein puts Twinfilin at the end of the strand. Thus, when Capping Protein is finally removed by CARMIL, Twinfilin carries on with disassembling the actin strands.

The tail of the Twinfilin protein looks like part of the CARMIL protein, suggesting that they might interact with Capping Protein in the same way. Attaching a fluorescent tag to the Twinfilin tail revealed that the two proteins compete to attach to the same part of the Capping Protein. When mouse cells produced extra Twinfilin, it blocked the effects of CARMIL, helping to grow the actin strands. V-1 attaches to Capping Protein in a different place, but Twinfilin was also able to interfere with its activity. When Twinfilin attached to the CARMIL binding site, it did not directly block V-1 binding, but it made the protein more likely to fall off.

Understanding how the actin cytoskeleton moves is a key question in cell biology, but it also has applications in medicine. Twinfilin plays a role in the spread of certain blood cancer cells, and in the formation of elaborate structures in the inner ear that help us hear. Understanding how Twinfilin and Capping Protein interact could open paths to new therapies for a range of medical conditions.
DOI: https://doi.org/10.7554/eLife.41313.002

(*Johnston et al., 2015*). More recently, it was shown that mouse Twinfilin isoforms accelerate barbed end depolymerization, similar to yeast Twinfilin, but do not induce robust pointed end depolymerization in conjunction with Srv2/CAP (*Hilton et al., 2018*). Collectively, these studies highlight the biological significance of Twinfilin.

The conserved barbed-end effects of Twinfilin are particularly interesting given that both yeast and mammalian Twinfilins bind to CP (*Falck et al., 2004*; *Palmgren et al., 2001*). Further, a barbed-end regulatory role for Twinfilin is suggested by its localization to the tips of stereocilia and filopodia, and to the barbed ends of *Drosophila* actin bristles (*Peng et al., 2009*; *Rzadzinska et al., 2009*; *Wahlström et al., 2001*). In addition, Twinfilin localizes to endocytic actin patches in yeast, and to lamellipodia and cell-cell junctions in animal cells (*Goode et al., 1998*; *Vartiainen et al., 2000*). Twinfilin's localization to cortical actin patches in yeast is dependent on its interaction with CP (*Palmgren et al., 2001*). In both yeast and mammals, this interaction is mediated by conserved sequences in the C-terminal tail region of Twinfilin (*Falck et al., 2004*). Despite the high affinity of the Twinfilin-CP interaction ($K_d$ ~10 nM for the yeast homologs [*Poukkula et al., 2011*]), studies have revealed no significant effects of Twinfilin on the barbed end capping activity of CP in vitro, and reciprocally, no obvious effect of CP on Twinfilin interactions with ADP-actin monomers (*Falck et al., 2004*). Thus, the functional significance of the Twinfilin-CP interaction has remained highly enigmatic.

CP is an obligate heterodimer, consisting of alpha and beta subunits, and binds stably to the barbed ends of actin filaments to block subunit addition and loss. CP is ubiquitous and highly

conserved across eukaryotes, and has universal roles in controlling the assembly of actin networks that drive cell morphogenesis and cell motility (*Cooper and Sept, 2008*; *Hart and Cooper, 1999*; *Mejillano et al., 2004*; *Schafer et al., 1994*; *Schafer et al., 1995*). In vitro, CP binds to the barbed ends of actin filaments with sub-nanomolar affinity, and dissociates from barbed ends very slowly (half-life of ~30 min) (*Schafer et al., 1996*). Given the relatively high abundance of CP in the cytosol (1–3 µM) and the strength of its interactions with barbed ends (*Cooper and Sept, 2008*), it is not surprising that cells have evolved a number of regulatory mechanisms to spatiotemporally restrict CP activity.

Cellular protein inhibitors of CP broadly fall into two classes: steric inhibitors and allosteric inhibitors. Steric inhibitors, which include V-1/myotrophin, bind to CP in a manner that physically obstructs its association with barbed ends (*Bhattacharya et al., 2006*; *Kim et al., 2007*; *Schafer et al., 1996*). V-1 is a highly abundant 13 kDa protein that binds CP with a $K_d$ ~40 nM and sterically blocks its ability to bind barbed ends (*Bhattacharya et al., 2006*; *Taoka et al., 2003*). Notably, however, V-1 does not catalyze dissociation of CP from barbed ends (*Bhattacharya et al., 2006*). In contrast, allosteric inhibitors induce conformational changes in CP that catalyze its dissociation from barbed ends ('uncapping' or 'displacing' CP), and also decrease but do not abolish its ability to bind barbed ends.

The major class of allosteric inhibitors is the capping protein interaction (CPI) motif family of proteins (*Edwards et al., 2014*). The founding and best characterized member of the CPI family is CARMIL (Capping Protein, ARP2/3 and Myosin I linker), which is conserved across metazoans (*Stark et al., 2017*). CARMIL catalyzes CP dissociation from barbed ends, reducing CP's affinity for barbed ends by ~100 fold, transforming it into a transient capper (*Fujiwara et al., 2014*; *Stark et al., 2017*; *Uruno et al., 2006*; *Yang et al., 2005*). CARMIL localizes to the leading-edge plasma membrane, where it promotes cell migration through direct interactions with CP (*Fujiwara et al., 2014*; *Liang et al., 2009*; *Stark et al., 2017*; *Yang et al., 2005*). Other proteins with CPI motifs include CD2AP, CKIP-1, CapZIP, CIN85, and WASHCAP (FAM21); their roles in regulating CP are less well understood. CPI-motif proteins share a common mode of interaction with CP, but are otherwise unrelated to each other (*Edwards et al., 2014*; *Hernandez-Valladares et al., 2010*). To date, binding partners of CP that antagonize its inhibitors, and thus function as 'pro-capping' factors, have not been reported.

Here, we uncover a novel role for Twinfilin in protecting CP from the negative regulatory effects of V-1 and CARMIL, and thus promoting actin filament capping. These and other data lead us to propose important revisions to current models for the CP regulatory cycle.

## Results

### CP inhibits mTwf1-mediated depolymerization by capping barbed ends

Because CP binding proteins have been studied predominantly in mammalian systems, we focused our investigation on mouse rather than yeast CP and Twinfilin. Mutagenesis on the yeast Twinfilin tail previously identified a mutant, *twf1-11,* that targets a cluster of positively charged residues (R328A, K329A, R330A, R331A) necessary for binding CP (*Falck et al., 2004*). While truncations of the C-terminal tail in mouse Twinfilin (mTwf1) also disrupt CP binding, the residues involved have not yet been defined. We therefore first sought to generate a specific mutant in mTwf1 that disrupts the interaction, analogous to yeast *twf1-11*. An alignment of the three mouse and three human Twinfilin isoforms, along with the single Twinfilin genes expressed in *S. cerevisiae* and *D. melanogaster* (*Figure 1A*), revealed a region that includes two of the basic residues mutated in the yeast *twf1-11* mutant. We mutated these two residues in mTwf1, changing them to alanines, to produce mTwf1-11 (K332A, R333A). To quantify binding of mTwf1 to CP, we performed fluorescence anisotropy assays using a mTwf1 tail peptide (317-350) labeled at its N-terminus with HiLyte488. The mTwf1 tail peptide displayed high affinity, concentration-dependent binding to CPα1β2, a major non-muscle isoform of CP in mammalian cells (*Figure 1B*). Moreover, full-length mTwf1 protein (unlabeled) competed with the labeled mTwf1 tail for CP binding, whereas full-length mTwf1-11 (unlabeled) did not (*Figure 1C*). Thus, the mTwf1-11 mutant effectively uncouples mTwf1 binding to CP.

Using mTwf1-11, we addressed how CP binding affects Twinfilin's actin depolymerization activities in total internal reflection fluorescence (TIRF) microscopy assays, by directly observing

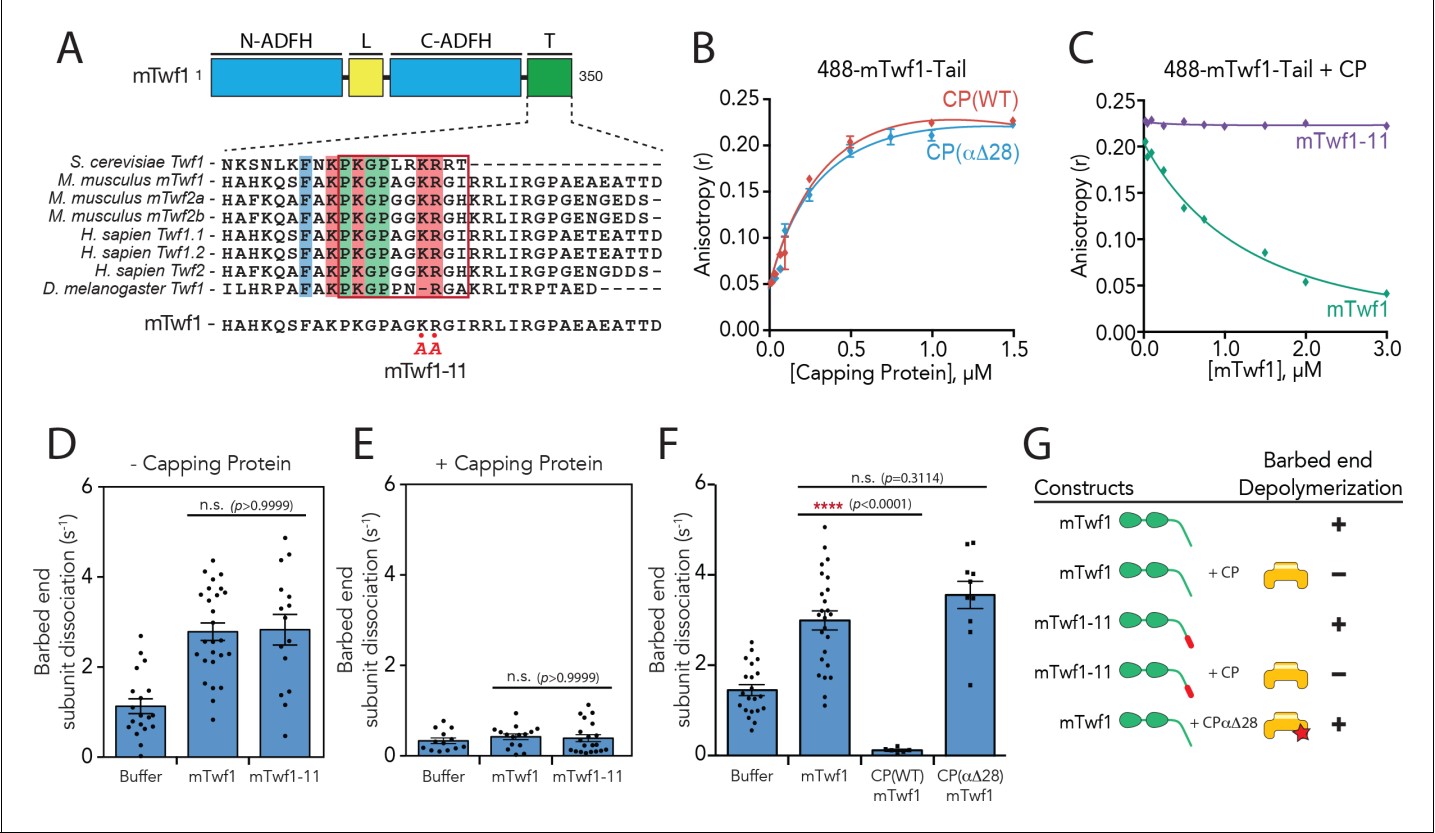

**Figure 1.** Barbed end capping by Capping Protein inhibits Twinfilin1-mediated depolymerization. (**A**) Mouse Twinfilin-1 (mTwf1) domain organization: ADF-H, actin depolymerization factor homology domain; L, linker; T, tail. Sequence alignment of tail regions of Twinfilin isoforms from different species with boxed region highlighting conservation of residues critical for binding to Capping Protein (CP). mTwf1-11 carries a mutation in the tail region (KR332,333AA) that disrupts binding to CP. (**B**) Fluorescence anisotropy measurement of 100 nM HiLyte488-labeled mTwf1 tail peptide mixed with increasing concentrations of the indicated CP construct. (**C**) Fluorescence anisotropy measurement of 100 nM HiLyte488-labeled mTwf1 tail peptide incubated in the presence 1 μM CP and increasing concentrations of either mTwf1 or mTwf1-11. Anisotropy values for each condition averaged from three independent experiments. (**D,E**) Rates of barbed end depolymerization (subunits $s^{-1}$) induced by 1 μM of the indicated mouse Twinfilin, in the (**D**) absence or (**E**) presence of 10 nM CP, determined from TIRF assays. Rates for each condition averaged from at least five filaments in each of two independent experiments. From left to right: (**D**) $n$ = 19, 26, and 15 and mean depolymerization rates 1.13, 2.784 and 2.81 subunits $s^{-1}$; (**E**) $n$ = 13, 15, and 20 and mean depolymerization rates 1.13, 2.784 and 2.81 subunits $s^{-1}$. (**F**) Rates of barbed end depolymerization (subunits $s^{-1}$) induced by 1 μM mTwf1, in the absence or presence of 1 μM of the indicated CP construct, determined from TIRF assays. Rates for each condition averaged from at least five filaments from at least one experiment. From left to right $n$ = 21, 25, 6, and 10; mean depolymerization rates 1.45, 2.991, 0.11, and 3.58 subunits $s^{-1}$. (**G**) Summary of barbed end depolymerization activity of mTwf1 constructs in combination with different CP constructs determined from TIRF assays (as in D,E,F). Error bars, s.e.m. ****$p\leq0.0001$, n.s. $p>0.05$ by one-way ANOVA with Tukey post hoc test.

DOI: https://doi.org/10.7554/eLife.41313.003

The following video is available for figure 1:

**Figure 1—video 1.** Supporting data for *Figure 1F*.

DOI: https://doi.org/10.7554/eLife.41313.015

depolymerization at actin filament barbed ends in real time. In agreement with previous observations using yeast and mouse Twinfilin (*Hilton et al., 2018*; *Johnston et al., 2015*), 1 μM mTwf1 accelerated barbed end depolymerization by 2–3 fold compared to control reactions (*Figure 1D*), and the addition of CP blocked this effect (*Figure 1E*). Further, mTwf1-11 exhibited a similar rate (*Figure 1D*), indicating that this mutant has wild type depolymerization activity, and thus separates Twinfilin's ability to bind CP from its ability to promote barbed-end depolymerization. Interestingly, the addition of CP was still able to block barbed-end depolymerization by mTwf1-11 (*Figure 1E*). These observations suggest that CP sterically blocks mTwf1 access to barbed ends, independent of its direct interaction with mTwf1. However, this left open the question of whether CP binding to mTwf1 might alter its mechanism of depolymerization independent of blocking the barbed end. To

address this possibility, we utilized a CP mutant, CPαΔ28, which truncates the C-terminal tentacle of the alpha subunit, severely inhibiting capping activity (*Kim et al., 2010*). Importantly, in binding assays the mTwf1 tail interacted equally well with wild-type CP and CPαΔ28, demonstrating that this mutant binds normally to mTwf1 (*Figure 1B*). In TIRF assays, equimolar amounts of CPαΔ28 did not significantly alter mTwf1 depolymerization activity (*Figure 1F*; *Figure 1—video 1*; also summarized in *Figure 1G*), suggesting that while CP blocks Twinfilin access to barbed ends, Twinfilin-CP direct interaction does not alter Twinfilin depolymerization activity.

## The twinfilin tail competes with CARMIL CPI motif for binding to CP

Given that CP binding does not affect Twinfilin's depolymerization activity, or other known activities of Twinfilin (*Falck et al., 2004*; *Johnston et al., 2015*; *Palmgren et al., 2001*), we next considered whether Twinfilin binding might influence CP functions in the presence of known regulators of CP. We were particularly interested in how Twinfilin might impact the regulation of CP by CPI-motif proteins such as CARMIL, since we noticed that the C-terminal tail regions of evolutionarily diverse Twinfilins share sequence homology with the CPI motifs of several CPI family proteins (*Figure 2A*). The consensus CPI motif is 17-amino acids long, with some additional contacts contributed from outside this motif, and tolerates significant divergence across the CPI-motif family (*Edwards et al., 2014*; *Hernandez-Valladares et al., 2010*). As an initial test, we used a mutant of CP, CP(RY), which alters two surface residues on the beta subunit (R15A, Y79A) that make essential contacts with CPI-motif proteins (*Edwards et al., 2014*; *Hernandez-Valladares et al., 2010*). The CP(RY) mutant is insensitive to inhibition and uncapping by CARMIL and disrupts binding with at least two other CPI-motif proteins, CD2AP and WASHCAP (FAM21) (*Edwards et al., 2015*). In fluorescence anisotropy binding assays, we observed that the CP(RY) mutant has approximately 20-fold reduced affinity for mTwf1 tail compared to wild type CP (*Figure 2B*). These data are consistent with mTwf1 and CPI-motif proteins sharing at least partially overlapping binding sites on CP. In addition, we asked whether introducing a mutation in the mTwf1 tail peptide at a conserved residue in CPI consensus sequences would alter binding to CP (Lys 325 in mTwf1; see red asterisk, sequence alignment in *Figure 2A*). In fluorescence anisotropy binding assays, we compared the abilities of wild-type and mutant (K325A) mTwf1 tail peptides to compete with labeled mTwf1 tail peptide for CP binding. This analysis revealed an ~30 fold reduction in binding affinity for the mutant (K325A) mTwf1 tail peptide compared to wild type peptide (*Figure 2C*).

We next asked whether the CP-binding region (CBR) of CARMIL1 (residues 964 – 1078) competes with mTwf1 tail for binding to CP. We observed that unlabeled CBR peptide competed with the fluorescent mTwf1 tail probe for CP binding (*Figure 2D*). These results indicate that CARMIL and mTwf1 directly compete for binding CP. Next, we more narrowly defined the region of CARMIL that competes with mTwf1 by using peptides that divide the CBR into its two conserved components, the CPI motif (969 – 1005) and the CARMIL-specific interaction (CSI) motif (1019 – 1037). The CSI makes additional contacts with CP, but is found only in CARMIL family members, and not in other CPI-motif proteins (*Edwards et al., 2014*). As expected based on Twinfilin's sequence similarity to CPI motifs, only the CPI-motif peptide and not the CSI peptide competed with mTwf1 tail for CP binding (*Figure 2D*). Together, these results suggest that Twinfilin is a divergent CPI-motif protein and has important implications for CP regulation in cells (see Discussion).

## Twinfilin attenuates CARMIL-mediated displacement of CP from barbed ends

Given that CARMIL and Twinfilin compete for binding to CP, we asked whether mTwf1 affects CARMIL's ability to displace CP from barbed ends. We addressed this question in pyrene actin assembly assays, where actin polymerization was initiated at time zero in the presence of CP and increasing concentrations of mTwf1, and after 400 s CARMIL1 CBR was spiked into the reaction. CARMIL1 alone (no mTwf1) strongly induced uncapping, leading to the rapid polymerization of previously-capped filament seeds (*Figure 3A*). However, increasing concentrations of mTwf1 attenuated CARMIL's uncapping effects (*Figure 3A*). These results are consistent with mTwf1 competing with CARMIL for binding CP, and thereby blocking uncapping.

To more directly observe mTwf1 effects on CARMIL-induced uncapping of barbed ends, we used TIRF microscopy. In these experiments, we used fluorescently labeled SNAP-tagged CP (SNAP-649-

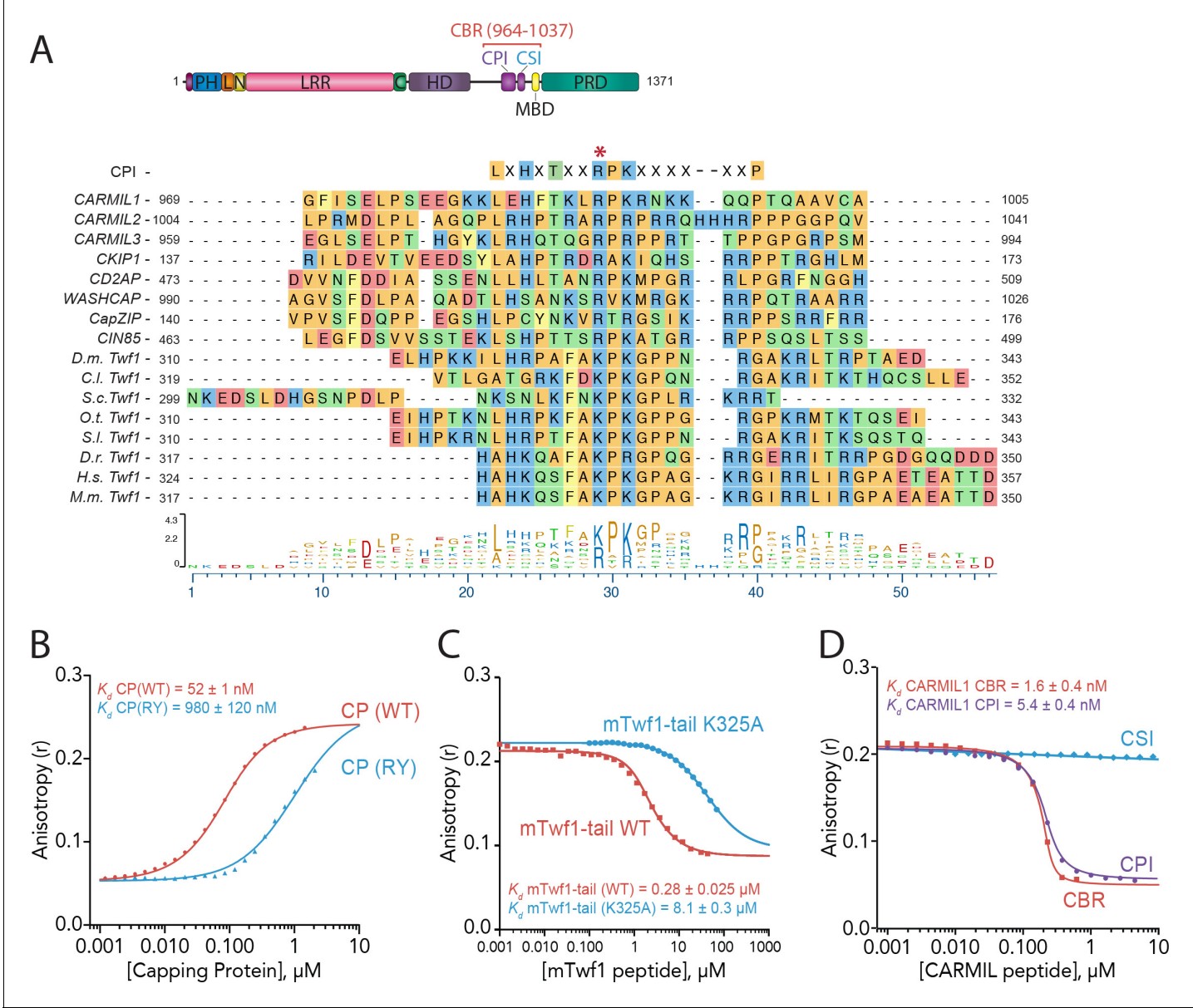

**Figure 2.** Twinfilin is a Capping Protein Interaction (CPI)-motif protein that competes with CARMIL for binding Capping Protein. (A) CARMIL domain organization: PH, pleckstrin-homology domain; L, linker; N-cap (N), LRR, leucine-rich repeat domain; C, C-cap; HD, helical dimerization domain; CBR, Capping Protein binding domain, consisting of CPI, Capping Protein interaction domain, and CSI, CARMIL-specific interaction sequence; MBD, membrane binding domain; PRD, proline-rich domain. Alignment between the Capping Protein Interaction (CPI) motif consensus sequence, and the CPI regions of *H. sapiens* (H.s.) CARMIL1 (UniProtKB Q5VZK9.1), CARMIL2 (UniProtKB Q6F5E8.2), CARMIL3 (UniProtKB Q8ND23.2), CKIP1 UniProtKB Q53GL0.2), CD2AP (CBI NP_036252.1), WASHCAP (Fam21) (UniProtKB Q9Y4E1.3), CapZIP (CBI NP_443094.3), CIN85 (UniProtKB Q96B97.2), and the tail sequences of Twinfilin homologs from *D. melanogaster* (D.m), *C. lectularius* (C.l.), *S. cerevisiae* (S.c.), *O. Taurus* (O.t.), *S. litura* (S.l.), *D. rerio* (D.r.), *H. sapiens* (H.s.), and *M. musculus* (M.m.). Twinfilin isoforms (*D.m.* Twf1 UniProtKB NP_650338, *C.l.* Twf1 UniProtKB XP_014258437.1, *S.c.* Twf1 GenBank GAX68393.1, *O.t.* Twf1 XP_022917989.1, *S.l.* Twf1 XP_022816377.1, *D.r.* Twf1 AAH67638.1, *H.s.* Twf1 UniProtKB NO_001229326.1, and *M.m.* Twf1 GenBank AAH15081.1). Amino acid color coding illustrates side chain chemistry similarities. The asterisk marks the residue we mutated in mTwf1 in panel. (C) The alignments were generated using the MAFFT algorithm in the DNASTAR Lasergene Suite/MegAlign Pro application (MegAlign Pro. Version 15.0. DNASTAR. Madison, WI.).(B) Fluorescence anisotropy measurement of 60 nM HiLyte488-labeled mTwf1 tail peptide mixed with increasing concentrations of the indicated CP construct. (C) Fluorescence anisotropy measurement of 40 nM TAMRA-labeled mTwf1 tail peptide incubated with 1 µM CP and different concentrations of wild type and mutant mTwf1 tail peptides. (D) Fluorescence anisotropy measurement of 60 nM HiLyte488-labeled mTwf1 tail peptide incubated in the presence of 240 nM CP and increasing concentrations of the indicated CARMIL fragment (CBR, CSI, or CPI). CSI failed to compete with HiLyte 488-mTwf1 tail peptide at the concentrations tested. Anisotropy values for each condition were averaged from three independent experiments.

DOI: https://doi.org/10.7554/eLife.41313.004

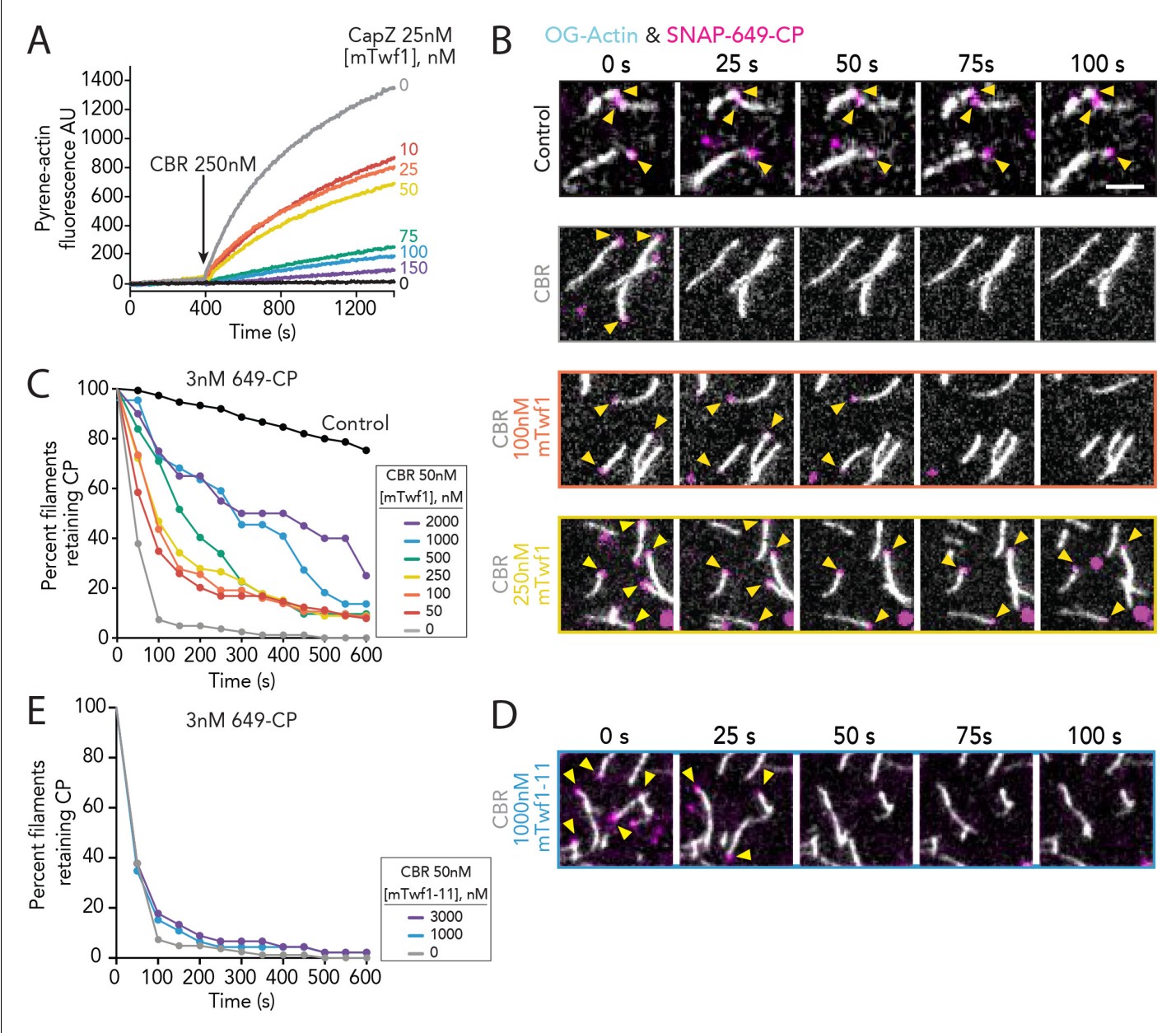

**Figure 3.** Direct interactions of Twinfilin with Capping Protein attenuate CARMIL-mediated uncapping. (**A**) Bulk fluorescence assays comparing the rates of actin assembly in the presence of 25 nM muscle Capping Protein (CPα1β1) and increasing concentrations of mTwf1. To initiate uncapping, 250 nM CBR fragment of CARMIL (see schematic, *Figure 2A*) was spiked into the reaction at 400 s. Data shown are representative curves from experiments repeated three independent times. (**B**) Representative time-lapse images from TIRF microscopy assays monitoring the displacement of labeled CP from barbed ends. Filaments were first polymerized and tethered using 1 μM actin (10% OG-labeled, 0.5% biotin–actin), then capped at their barbed ends by flowing in SNAP-649-CP (100% labeled). Next, 50 nM CBR fragment of CARMIL and different concentrations of mTwf1 were flowed in, and CP dissociation was monitored over time. Scale bar, 5 μm. (**C**) Quantification of the percentage of filaments retaining CP at the barbed ends in the presence of 50 nM CBR fragment of CARMIL and variable concentrations of mTwf1, determined from TIRF reactions as in (**B**). Control curve, buffer alone (no CBR or mTwf1). *n* > 45 events measured from at least two independent experiments. (**D**) Representative time-lapse images from TIRF microscopy assays monitoring CP displacement from barbed ends, analyzed as in (**B**), except using 1 μM mTwf1-11 instead of mTwf1. *n* > 45 events measured from at least two independent experiments. (**E**) Quantification of the percentage of filaments retaining CP at the barbed end in the presence of 50 nM CBR fragment of CARMIL and different concentrations of mTwf1-11, determined from TIRF assays as in (**D**). *n* > 45 events measured from at least two independent experiments.

DOI: https://doi.org/10.7554/eLife.41313.005

The following figure supplement is available for figure 3:

**Figure supplement 1.** Supporting data for *Figure 3* showing that multiple Twinfilin isoforms antagonize CARMIL uncapping of barbed ends.

*Figure 3 continued on next page*

*Figure 3 continued*

DOI: https://doi.org/10.7554/eLife.41313.006

CP; 100% labeled) to monitor lifetimes of CP molecules on filament barbed ends (*Bombardier et al., 2015*). Filaments were first polymerized to a desired length (~10 μm) and then capped by flowing in SNAP-649-CP. Free CP was washed out, and then proteins of interest (or control buffer) were flowed in. Capped filaments were identified in the field of view prior to flow-in, and then monitored after flow-in to measure the dwell time of SNAP-649-CP. As expected, in the absence of other factors, SNAP-649-CP had a long dwell time, remaining on barbed ends for tens of minutes (*Figure 3B and C*). However, when CARMIL1 CBR was introduced, this led to the rapid displacement of SNAP-649-CP, with complete loss of CP from barbed ends by 100 s (*Figure 3B and C*). The addition of mTwf1 with CARMIL1 CBR attenuated the uncapping effects in a concentration-dependent manner (*Figure 3B and C*). Further, this attenuation required direct interactions between Twinfilin and CP, as mTwf1-11 failed to protect CP from CARMIL uncapping (*Figure 3D and E* and *Figure 3—figure supplement 1*). Similar effects were observed for the other major isoform of mouse Twinfilin that is expressed in non-muscle cells, mTwf2a (*Figure 3—figure supplement 1*) (*Nevalainen et al., 2011*; *Vartiainen et al., 2003*).

## Twinfilin accelerates the dissociation of V-1 from CP

We next considered whether Twinfilin binding to CP might affect the activities of CP inhibitor V-1/myotrophin, which is distinct from CPI-motif proteins in its mode of CP interaction. Unlike CARMIL, V-1 does not displace CP from barbed ends; instead, it sequesters CP and blocks it from binding filament ends (*Bhattacharya et al., 2006*; *Jung et al., 2016*; *Taoka et al., 2003*). In contrast to the CARMIL binding site on CP, which partially encircles the 'stalk' of the CP heterodimer (*Hernandez-Valladares et al., 2010*; *Johnson et al., 2018*; *Zwolak et al., 2010*), V-1 interacts with CP on the opposite face, sterically blocking binding to the filament end (*Johnson et al., 2018*; *Takeda et al., 2010*; *Zwolak et al., 2010*). To test how Twinfilin might affect the interaction of CP with V-1, we used pyrene-actin seeded elongation assays (*Figure 4A*). As expected, filament seeds pre-incubated with CP and then mixed with pyrene-actin monomers displayed minimal growth, whereas the addition of V-1 restored actin assembly to uncapped levels. Somewhat to our surprise, the further addition of mTwf1 suppressed V-1's effects, restoring capping activity, while mTwf1-11 had no effect (*Figure 4A and B*). These effects were unexpected given the above-mentioned differences in Twinfilin's predicted and V-1's known binding sites on CP, and our observation that even high concentrations of V-1 (1000-fold excess to mTwf1 tail probe) fail to compete with mTwf1 for CP binding in anisotropy assays (*Figure 4C*). These results suggest that mTwf1 attenuates V-1 effects on CP via an allosteric mechanism, distinct from a simple steric binding competition.

In probing the mechanism further, we drew inspiration from a study by Fujiwara and colleagues, showing that CARMIL forms a transient ternary complex with V-1 and CP, leading to accelerated dissociation of V-1 from CP (*Fujiwara et al., 2014*). We asked whether mTwf1 might similarly catalyze the dissociation of V-1 from CP. In stopped-flow fluorescence assays, fluorescently labeled V-1 (TAMRA-V-1) was first allowed to bind CP, and then mixed at time zero with an excess of unlabeled V-1. The resulting decrease in fluorescence reflects the spontaneous dissociation of TAMRA-V-1 from CP (*Figure 4D*). The rate of V-1 dissociation from CP increased in the presence of increasing concentrations of mTwf1, pointing to the possible formation of a transient ternary complex that destabilizes V-1 interactions with CP (*Figure 4D and E*). Importantly, mTwf1-11 failed to enhance V-1 dissociation (*Figure 4E*), showing that this effect depends on direct interactions between mTwf1 tail and CP. These results demonstrate that CARMIL and Twinfilin share a common function in catalyzing the dissociation of V-1 from CP using their CPI motifs to bind CP, despite having different effects on the displacement of CP from barbed ends.

## Structural evidence for the twinfilin tail interacting with the CPI-binding site on CP

Given the observed competition between mTwf1 tail peptide and the CPI motif of CARMIL for binding to CP, and the similarity between mTwf1 and CARMIL in catalyzing V-1 dissociation from CP, we

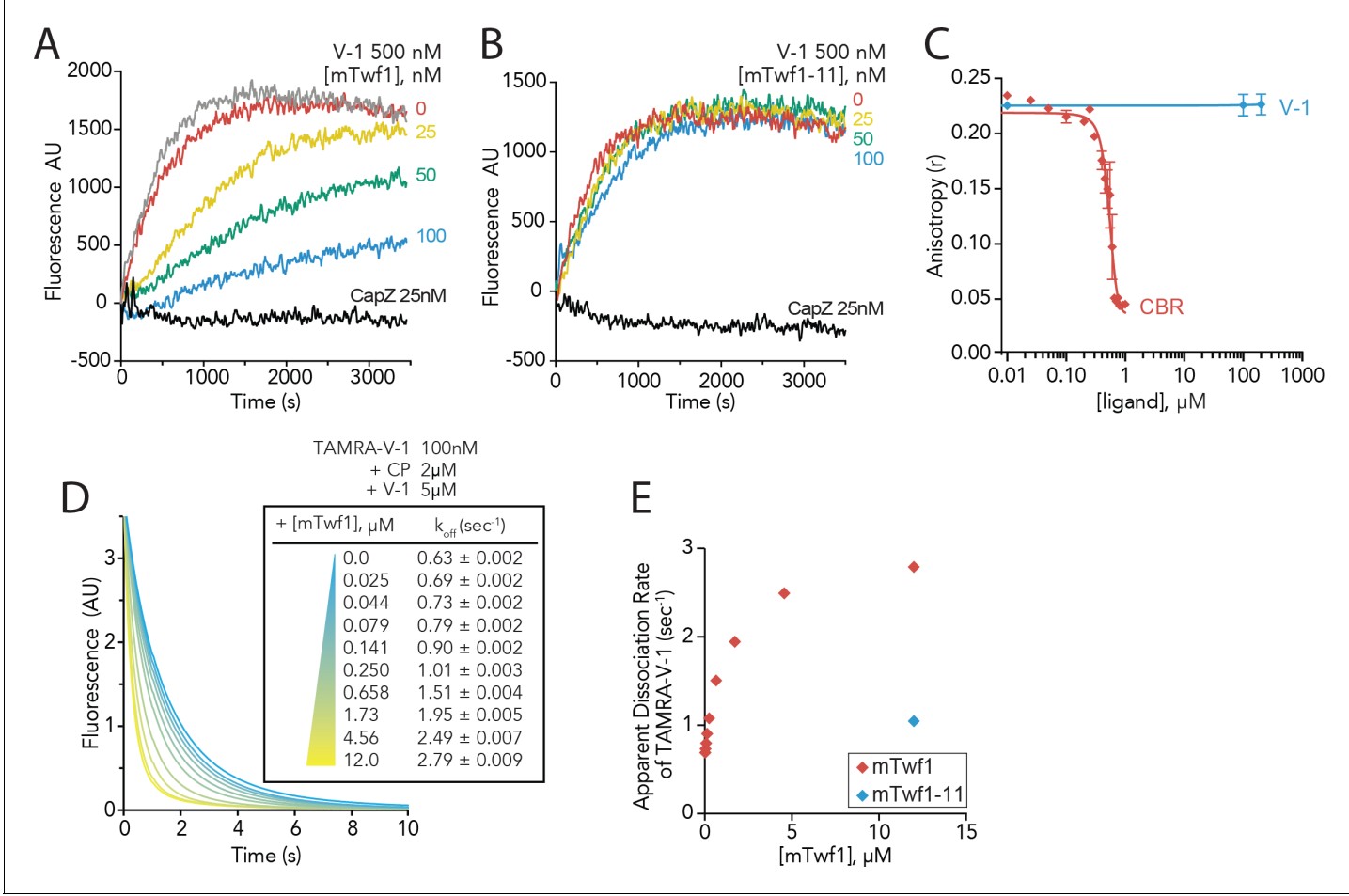

**Figure 4.** Twinfilin's direct binding to Capping Protein accelerates the disassociation of V-1 to promote capping of filaments. (**A, B**) Seeded elongation assays comparing the rates of actin assembly from spectrin-F-actin seeds (grey) in the presence of 0.5 μM actin (10% pyrene-labeled), 25 nM muscle Capping Protein (CapZ), 500 nM V-1, and variable concentrations of mTwf1 (**A**) or mTwf1-11 (**B**) as indicated. Data shown are representative curves from experiments performed three independent times. (**C**) Fluorescence anisotropy measurement of 100 nM HiLyte488-labeled mTwf1 tail peptide mixed with 1 μM mouse Capping Protein (CP) and variable concentrations of CBR fragment of CARMIL or V-1. Rates for each condition averaged from three independent experiments. (**D**) Stopped-flow fluorescence assays measuring the kinetics of dissociation of 50 nM TAMRA-V-1 from 1 μM CP upon addition at time zero of 2.5 μM unlabeled V-1 and variable concentrations of mTwf1 as indicated. Apparent dissociation rates are listed for each condition. (**E**) Apparent dissociation rates of TAMRA-V-1 for different concentrations of mTwf1 are from (**D**); and for 12 μM mTwf1−11 = $1.0 \pm 0.003$ s$^{-1}$. Anisotropy values for each condition were averaged from five independent experiments.

DOI: https://doi.org/10.7554/eLife.41313.007

sought structural evidence for the nature of the interaction between mTwf1 and CP. We hypothesized that the binding sites for mTwf1 and the CPI motif were likely overlapping. To test this hypothesis, we used hydrogen-deuterium exchange with mass spectrometry (HDX-MS) to interrogate the conformational dynamics and solvent accessibility of the backbone and sidechains of CP, free and in complex with Twf1. Further, we compared our results to those in our recent study on the interactions of CARMIL with CP using the same approach (*Johnson et al., 2018*). We tested three different forms of mTwf1: a short tail peptide (residues 317–350), a longer tail peptide (residues 305–350), and full-length mTwf1. These constructs were added to CP, either full-length alpha/beta heterodimer, or full-length alpha subunit with a beta subunit truncated at its C-terminus, removing the actin-binding beta tentacle. The results were essentially the same in each case. The presence of mTwf1 resulted in protection from H-D exchange at the N-terminal stalk of CP (*Figure 5A*, *Figure 5—figure supplements 1* and *2*). Similar effects to H-D exchange were observed upon CARMIL binding to CP (*Johnson et al., 2018*); also shown here in *Figure 5B*), which correspond well with the CPI-motif binding site defined by X-ray crystallography and solution NMR studies (*Hernandez-*

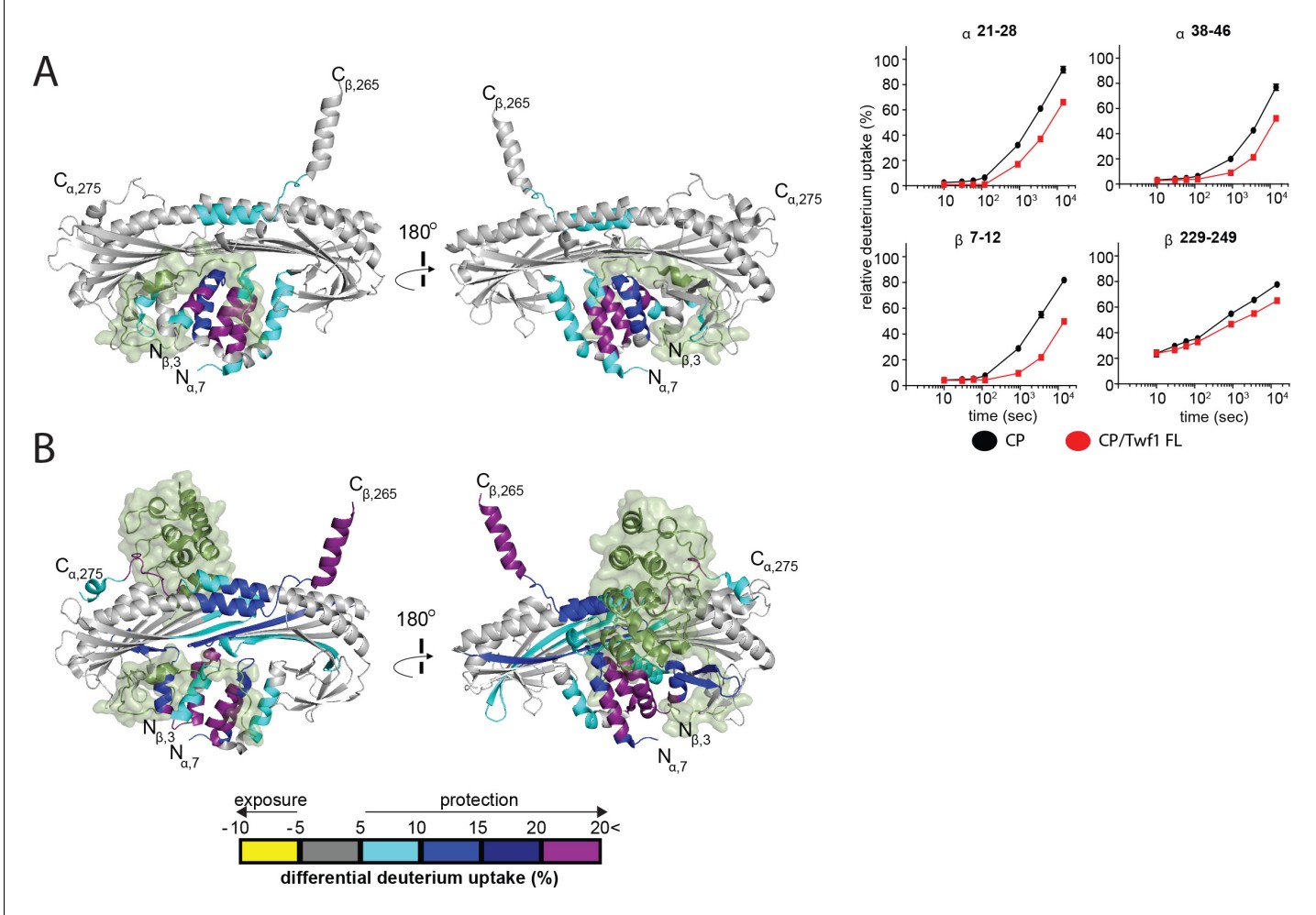

**Figure 5.** HDX-MS analysis of Twinfilin reveals effects on Capping Protein structure near the CPI motif-binding site. (**A**) A cartoon representation of a crystal structure of CP, based on PDB 3AAA (*Takeda et al., 2010*). Differences in deuterium uptake induced by mTwf1 binding to CP are displayed as a color gradient (see scale at bottom of panel (**B**) CPI domain of CARMIL overlaid on to its binding site on CP (around the stalk). Representative comparisons of deuterium uptake curves for free CP (black) with mTwf1 bound CP (red) for CP alpha subunit (upper panels) and CP beta subunit (lower panels). Error bars representing the results of *t*-tests between samples are shown above each time point to illustrate statistical significance. When error bars are not shown explicitly, the error is within the radius of the symbol. Data shown are representative curves from experiments repeated two independent times. (**B**) A cartoon representation of a crystal structure of CP, showing the differences in deuterium uptake induced by CBR domain of CARMIL binding to CP are displayed as a color gradient (see scale at the bottom). CPI domain of CARMIL overlaid on to its binding site on CP (around the stalk), V-1 is overlaid on its binding site on CP (barbed end binding surface) for comparison.

DOI: https://doi.org/10.7554/eLife.41313.008

The following figure supplements are available for figure 5:

**Figure supplement 1.** Supporting data for *Figure 5* showing differential HDX results.
DOI: https://doi.org/10.7554/eLife.41313.009

**Figure supplement 2.** Supporting data for *Figure 5* showing differential HDX results.
DOI: https://doi.org/10.7554/eLife.41313.010

*Valladares et al., 2010*; *Takeda et al., 2010*; *Zwolak et al., 2010*). For mTwf1, we also observed H-D exchange protection of a small region on CP corresponding to the V-1 binding site (*Figure 5A and B*, *Figure 5—figure supplements 1* and *2*), consistent with our results described above for the effects of mTwf1 in promoting V-1 dissociation from CP. These structural effects are also consistent with our previous results for CARMIL, which alters the V-1 binding site (*Johnson et al., 2018*). However, it is worth noting that mTwf1-induced changes in CP conformation at the actin-binding

interface were not as extensive as those induced by CARMIL, which is consistent with CARMIL, but not mTwf1, weakening CP binding to actin at the barbed ends.

## Twinfilin and CP colocalize in cells and have similar knockdown phenotypes

To investigate the functional relationship between Twinfilin and CP in cells, we started by asking whether mTwf1 and CP colocalize. While Twinfilin and CP have been localized individually, and are each reported to be enriched at the tips of filopodia and stereocilia, endocytic actin patches, lamellipodia, and *Drosophila* bristles (*Avenarius et al., 2017*; *Falck et al., 2004*; *Goode et al., 1998*; *Nevalainen et al., 2011*; *Peng et al., 2009*; *Rzadzinska et al., 2009*; *Sinnar et al., 2014*; *Vartiainen et al., 2000*), to our knowledge they have never been co-imaged in vertebrate cells. To address this, we performed immunofluorescence on CP and Twinfilin in mouse B16F10 melanoma cells, co-staining the cells with Alexa 568-phalloidin to visualize F-actin. We observed strong colocalization of Twinfilin and CP throughout the cell and a co-enrichment at the actin-rich leading and trailing edges (*Figure 6A and B*). Further, quantitative western blotting showed that Twinfilin and CP are present at ~1:2 molar ratio in B16F10 cells (*Figure 6C*, *Figure 6—figure supplement 1*). Previous studies reported the concentration of CP in B16F10 cells to be ~1 µM (*Fujiwara et al., 2014*; *Pollard and Borisy, 2003*), suggesting that mTwf1 is present at ~0.5 µM. Given the high affinity of the Twinfilin-CP interaction ($K_d$ = 50 nM), these observations are consistent with mTwf1 being associated with a substantial fraction of the CP in cells.

The ability of Twinfilin to function as a 'pro-capping' factor in vitro, by antagonizing the inhibitory effects of V-1 on CP, predicted that genetic loss of mTwf1 might at least partially phenocopy loss of CP. While a number of studies have examined how Twinfilin mutations affect whole animal development and physiology (*Iwasa and Mullins, 2007*; *Meacham et al., 2009*; *Nevalainen et al., 2011*; *Wahlström et al., 2001*; *Wang et al., 2010*; *Yamada et al., 2007*), we are unaware of any studies that have investigated how loss of Twf1 affects the morphology and actin organization of cultured mammalian cells. Using RNAi silencing in B16F10 cells, we separately depleted endogenous mTwf1 and CP, which was verified by both western blotting (*Figure 6E and F*) and immunostaining (*Figure 6—figure supplement 1*). Knockdown of either mTwf1 or CP led to a similar, marked increase in the density of peripheral protrusions or microspikes with a concomitant loss of lamellipodial surfaces (*Figure 6F and G*). Similar phenotypes have been reported for CP depletion in multiple cell lines (*Edwards et al., 2013*; *Edwards et al., 2015*; *Mejillano et al., 2004*; *Sinnar et al., 2014*). Expression of an RNAi-refractive mTwf1 construct, but not mTwf1-11, rescued the defects caused by depletion of endogenous mTwf1 (*Figure 6F and G*; *Figure 6—figure supplement 1*), demonstrating that these cellular functions of mTwf1 critically depend on its interaction with CP.

We also made the unexpected observation that knockdown of CP was accompanied by a dramatic reduction in Twinfilin levels in cells, as seen by both western blotting (*Figure 6D*) and immunofluorescence (*Figure 6—figure supplement 1*). This effect was confirmed using a second RNAi oligonucleotide that targets a different region of CP (siCP2, *Figure 6D*). Further, it was observed in additional cell lines besides B16F10, including Neuro-2A and NIH-3T3 cells (*Figure 6—figure supplement 1*). These observations support the closely intertwined relationship of CP and Twinfilin in vivo.

Our results above also call into question whether the full extent of the phenotype caused by knockdown of CP (*Figure 6G*) is due to loss of CP, or instead is partly due to the accompanying loss of Twinfilin. To address this, we restored mTwf1 levels in cells depleted of CP by driving mTwf1 expression from a rescue plasmid, which was confirmed by western blotting and immunofluorescence (*Figure 6—figure supplement 1*). Forced expression of mTwf1 partially rescued the defects associated with CP depletion, indicating that a portion of the original defects observed after CP knockdown were likely due to the accompanying loss of mTwf1. These observations also suggest that many previously reported phenotypes arising from CP knockouts and knockdowns should be revisited or reinterpreted with the potential loss of Twinfilin in mind.

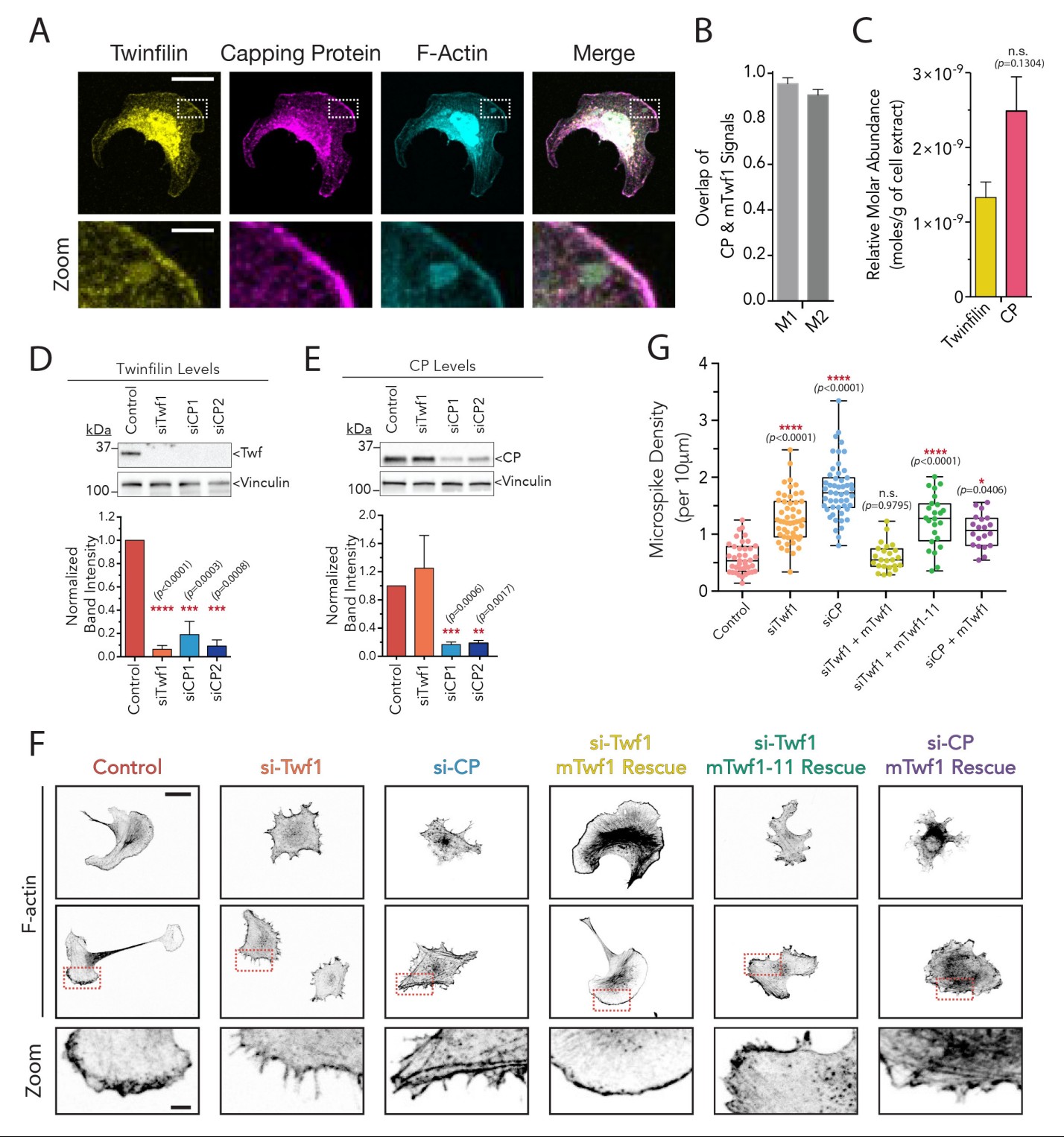

**Figure 6.** mTwf1 and Capping Protein colocalize and have similar phenotypes in B16F10 melanoma cells. (**A**) Representative images from immunofluorescence staining showing colocalization of endogenous mTwf1 (yellow) and Capping Protein (magenta). Scale bar, 20 μm. Close ups of boxed regions shown in Zooms; scale bar, 4 μm. (**B**) Mander's correlation coefficient (**M1 and M2**) values of overlap between mTwf1 and Capping Protein (CP) measured from cells ($n$ = 67 cells) as in (**A**). Error bars, s.e.m. (**C**) Comparison of the relative abundance of mTwf1 and Capping Protein (CP) in B16F10 cells measured from western blot analysis. Data averaged from four separate experiments. Error bars, s.d. n.s. p>0.05 by t-test. (**D,E**) Representative western blots and quantification of cellular levels of mTwf1 (**D**) and CP (**E**) in B16F10 cells treated with siRNA against mTwf1 (si-Twf1) or

*Figure 6 continued on next page*

*Figure 6 continued*

CP (si-CP) or negative control (Control). Band intensity for control cells was set to 1.0. Data averaged from at least three separate experiments., error bars, s.d. (F) Representative images showing F-actin immunofluorescence in B16F10 cells treated with siRNA against mTwf1 (si-Twf1) or CP (si-CP) or negative control (Control); siRNA treated cells (si-Twf1 or si-CP) were also rescued using plasmids expressing si-resistant FL-myc-mTwf1 (WT or mTwf1-11). Scale bar, 20 μm. Close ups of boxed regions shown in Zooms; scale bar, 4 μm. (G) Microspike density in cells treated as in (D). Box and whisker plots show mean, first and third quartile, and the maximum and minimum values. Data averaged from two experiments. From Left to right: $n$ = 45, 53, 51, 24, 24, and 20 and mean microspike density 0.69, 1.34, 1.77, 0.59, 1.24, and 1.01 filopodia per 10 μm of cell cortex. Error bars, s.e.m. ****$p \leq 0.0001$, *$p \leq 0.05$, n.s. $p > 0.05$ by one-way ANOVA with Tukey post hoc test.

DOI: https://doi.org/10.7554/eLife.41313.011

The following figure supplement is available for figure 6:

**Figure supplement 1.** Supporting data for *Figure 6*.

DOI: https://doi.org/10.7554/eLife.41313.012

## Defects caused by CARMIL1 hyperactivity can be rescued by elevated twinfilin expression

Finally, we tested the prediction of our biochemical observations that loss of capping activity in cells caused by overexpressed CARMIL1 should be restored by co-overexpression of Twf1. B16F10 cells ectopically expressing CARMIL1 showed morphological defects similar to loss of CP, and ectopic mTwf1 expression rescued the defects (*Figure 7A and B*). Importantly, ectopic expression of mTwf1 alone caused no significant change in cell morphology. These results support our biochemical observations, and suggest that Twf1 promotes capping in vivo, at least in part by competing with CARMIL for CP binding and antagonizing the uncapping effects of CARMIL.

## Discussion

Twinfilin and CP have been inextricably linked as interacting partners in yeast and animal cells for over 15 years (*Palmgren et al., 2001*), yet until now it has remained a mystery what function their interaction serves. Here we discovered that Twinfilin binds to CP using an orphan CPI-like sequence in its C-terminal tail region, and through this interaction protects CP from inhibition and/or barbed end displacement by CARMIL and V-1. We found that Twinfilin binds to CP in a competitive manner with the CPI motif of CARMIL, interacts with a site on CP similar to that of CARMIL, and attenuates CARMIL-mediated uncapping of actin filaments. Separately, Twinfilin binding to CP also accelerates V-1 dissociation from CP, despite Twinfilin and V-1 having non-overlapping binding sites on CP. This might be achieved by an allosteric mechanism, given that CARMIL uses its CPI motif to induce V-1 dissociation from CP through allosteric changes (*Fujiwara et al., 2014*; *Johnson et al., 2018*). Thus, we have demonstrated that Twinfilin promotes capping by protecting CP from interactions with V-1 and CARMIL. This functional role for Twinfilin is further supported in vivo by our observations of: (i) strong colocalization of Twinfilin and CP, (ii) knockdowns of mTwf1 and CP that each give rise to similar defects in cell morphology, and (iii) over-expression of mTwf1 suppressing defects caused by CARMIL hyperactivity. Taken together, these results reveal that Twinfilin is a new member of the CPI-motif family of proteins, and the first within this group to show the ability to bind CP without reducing CP affinity for barbed ends, and antagonize the negative regulatory effects of another CPI protein.

These functions of Twinfilin provide important new insights into the CP regulatory cycle. The best working model to date has been the Fujiwara model (*Fujiwara et al., 2014*) (depicted here as 'Earlier Model'; *Figure 7C*). It posits that the majority of CP in the cytosol is bound to V-1, in an inactive state, which then can be locally 'activated' by CARMIL at the leading edge. However, a caveat to this model is that it suggests CP-CARMIL complexes are the dominant capping species in the cell, despite this complex having ~100 fold reduced affinity for barbed ends compared to free CP. While this could potentially explain dynamic capping and uncapping near the plasma membrane, consistent with GFP-CP single molecule speckle analysis (*Miyoshi et al., 2006*), it does not explain how cells maintain a pool of 'capping competent' CP further back from the leading edge, where CP is needed to cap barbed ends in stress fibers and other actin networks, and may cap barbed ends generated by severing to promote filament disassembly. This model goes on to suggest that an

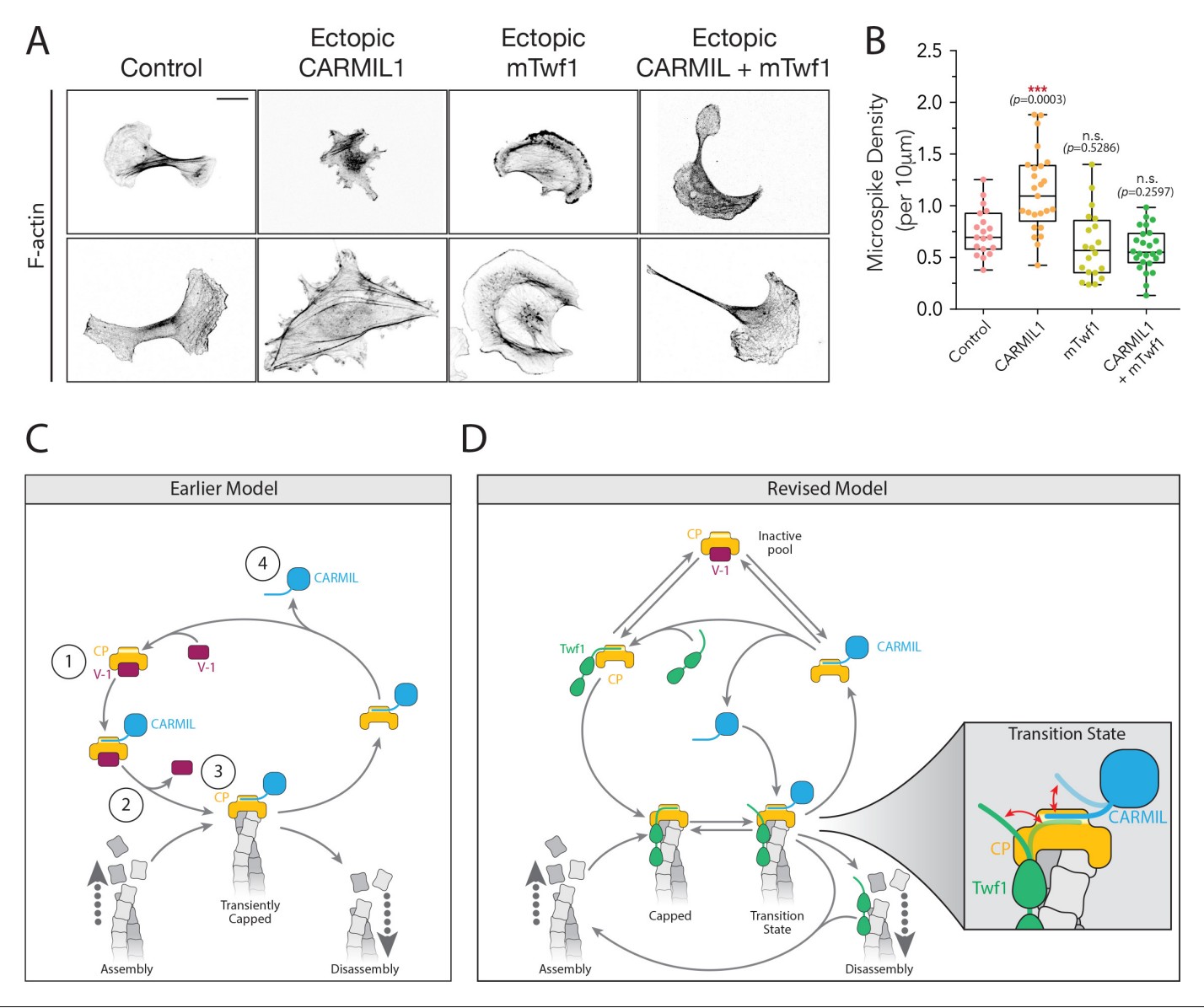

**Figure 7.** Overexpression of Twinfilin suppresses morphological defects caused by CARMIL hyperactivity. (A) Representative images of F-actin staining in untreated B16F10 cells (control), and cells transfected with Flag-CARMIL1, full-length (FL)-myc-mTwf1, or both. Scale bar, 20 μm. (B) Average Microspike density in cells treated as in (A). Box and whisker plots show mean, first and third quartile, and the maximum and minimum values. Data averaged from two experiments (n = 19–25 cells per condition). Data averaged from two experiments. From Left to right: *n* = 19, 25, 20, and 25; mean microspike density 0.75, 1.13, 0.62, 0.58 filopodia per 10 μm of cell cortex. Error bars, s.e.m. ***p≤0.001, n.s. p>0.05 by one-way ANOVA with Tukey post hoc test. (C) 'Earlier' model for CP regulatory cycle, adapted from Fujiwara and colleagues (*Fujiwara et al., 2014*). Proposed steps in model: (1) V-1 globally inhibits Capping Protein (CP) in the cytoplasm, (2) membrane-associated CARMIL (at the protruding cell edge) catalyzes dissociation of V-1 from CP, (3) the resulting CARMIL-CP complex is partially active, binding weakly to free barbed ends to provide capping function, (4) an unknown factor or mechanism promotes dissociation of CARMIL from CP, allowing V-1 to rebind CP and complete the cycle. (D) Our revised working model for the CP regulatory cycle. We propose that V-1 functions to maintain a cytosolic reservoir of inactive CP, from which Twinfilin and CARMIL activate CP, generating two distinct forms of active CP in cells: Twinfilin-CP complexes and CARMIL-CP complexes. Twinfilin-CP complexes are fully active and support stable capping of barbed ends. In contrast, CARMIL-CP complexes have ~100 fold reduced affinity for barbed ends, and may therefore more transiently cap barbed ends, permitting restricted network growth at the cell membrane where CARMIL localizes. CARMIL and Twinfilin directly compete with each other for binding CP (shown in close up of Transition state), which may result in the displacement of CP from Twinfilin. This would leave Twinfilin at the barbed end to catalyze depolymerization, or alternatively return filaments back to the original state of assembly.
DOI: https://doi.org/10.7554/eLife.41313.013

The following figure supplement is available for figure 7:

**Figure supplement 1.** Structural model for a ternary complex formed by Twinfilin, Capping Protein and the barbed end of an actin filament.

*Figure 7 continued on next page*

*Figure 7 continued*

DOI: https://doi.org/10.7554/eLife.41313.014

unknown factor or mechanism dissociates the CP-CARMIL complex, allowing V-1 to rebind CP, restoring it to an inactive state.

In light of our results, we propose several additions and revisions to the Fujiwara model (see 'Revised Model'; *Figure 7D*). First, we suggest that Twinfilin's protective effects on CP, in particular against V-1, allow cells to maintain a larger pool of fully active CP (Twinfilin-CP complexes) in the cytosol than was previously thought. This view is supported by the relatively high abundance of Twinfilin in cells (~0.5 µM, compared to ~1 µM CP) its high affinity for CP ($K_d$ = 50 nM), and its ability to increase the rate of dissociation of V-1 from CP. Given these observations, we propose that a substantial fraction of CP is available in a fully active state, as Twinfilin-CP complexes, even in the presence of a high concentration of V-1 in the cytosol (~3 µM) (*Fujiwara et al., 2014*; *Pollard and Borisy, 2003*). Second, we propose that V-1 functions to maintain a cytosolic reservoir of inactive CP, mobilized by Twinfilin and/or CARMIL dissociating V-1 to generate 'stable capping' (Twinfilin-CP) in the cytosol and possibly 'transient capping' (CARMIL-CP) complexes at the plasma membrane, respectively. CARMIL-CP complexes at the plasma membrane could facilitate actin network growth to drive leading edge protrusion. In contrast, Twinfilin-CP complexes in the cytosol may facilitate stable capping of barbed ends to limit network growth and promote filament disassembly and turnover. Third, we propose that the association of Twinfilin-CP complexes with barbed ends primes filaments for disassembly. Our data show that CARMIL, and/or other CPI proteins, compete with Twinfilin for binding CP. These interactions may competitively remove CP from barbed ends, leaving Twinfilin at the barbed end to processively depolymerize filaments, either alone or in combination with Srv2/CAP (as depicted in *Figure 6D*) (*Hilton et al., 2018*; *Johnston et al., 2015*). In this manner, the interaction of Twinfilin with CP could serve not only to initially promote capping, and thus limit network growth, but also to position Twinfilin at barbed ends for subsequently catalyzing the disassembly of filaments.

In summary, our results show that functions of mammalian Twinfilin and CP are closely intertwined. This functional relationship is likely to extend to other species given CPI motif sequence conservation in the Twinfilin tail region (*Figure 2A*) and the conserved nature of the Twinfilin-CP interaction. Indeed, *S. cerevisiae* Aim21 was recently identified as the first yeast CPI motif-containing protein, and was shown to regulate CP function at cortical actin patches (*Farrell et al., 2017*; *Shin et al., 2018*). We generated a structural model to explore the possible ternary complex formed by Twinfilin, CP, and the barbed end of an actin filament (*Figure 7—figure supplement 1*). In this model, the Twinfilin tail is long enough to allow for simultaneous binding of Twinfilin's CPI motif to CP and Twinfilin's C-terminal ADFH domain to an actin subunit at the barbed end. Further, there are no clashes in binding between CP and Twinfilin on actin. It is worth noting that CP and Twinfilin appear to be able to associate with barbed ends individually or as a CP-Twinfilin complex, but with distinct consequences for the function and dynamics of actin networks. CP alone stably caps barbed ends, blocking subunit addition or loss, and our results suggest that CP-Twinfilin complexes may do the same. However, when Twinfilin alone associates with barbed ends, it drives processive depolymerization, while blocking new assembly (*Hilton et al., 2018*; *Johnston et al., 2015*). Thus, despite key differences in the nature of their associations with barbed ends, CP and Twinfilin each inhibit filament growth, likely explaining why Twinfilin can replace CP in reconstituted actin motility assays in vitro (*Helfer et al., 2006*).

Finally, our data add to a broader emerging view that actin dynamics in vivo are controlled by a complex set of barbed end-associated factors, many of which interact with each other and/or stimulate each other's dissociation from barbed ends. These multi-component mechanisms may allow cells to control rapid transitions at filament ends through different functional states, including (i) formin-bound elongation, (ii) paused growth by formin-CP 'decision complexes', (iii) stable or transiently capped states by CP alone or CP-Twinfilin complexes (*Bombardier et al., 2015*; *Shekhar et al., 2015*), and (iv) depolymerization by Twinfilin, Cofilin, and/or Srv2/CAP. These molecular mechanisms for regulating barbed end growth are vastly more elaborate and dynamic than once thought, and help explain the exquisite spatiotemporal control that cells have in tuning actin network dynamics.

# Materials and methods

**Key resources table**

| Reagent type (species) or resource | Designation | Source or reference | Identifiers | Additional information |
|---|---|---|---|---|
| Antibody | Rabbit anti-Twinfilin | Pekka Lappalainen (Univ. Helsinki) | | WB (1:1000) IF (1:100) |
| Antibody | mouse anti-Capping Protein | Development Studies Hybridoma Bank | Cat: 3F2 | WB (1:2000) IF (1:50) |
| Antibody | Mouse anti-Flag | Sigma Aldrich | Cat: F3165 | WB (1:5000) IF (1:500) |
| Antibody | Rabbit anti-Myc | GeneTex | Cat: GTX29106 | WB (1:5000) IF (1:500) |
| Antibody | Goat anti-mouse-HRP | GE Healthcare | Cat: 31430 | WB (1:10000) |
| Antibody | Goat anti-rabbit-HRP | GE Health care | Cat: 31460 | WB (1:10000) |
| Antibody | Donkey anti-rabbit Alexa Flour 488 | Thermo Fisher Scientific | Cat: A21206 | IF (1:1000) |
| Antibody | Donkey anti-mouse Alexa Flour 488 | Thermo Fisher Scientific | Cat: A21202 | IF (1:1000) |
| Antibody | Goat anti-rabbit Alexa Flour 633 | Thermo Fisher Scientific | Cat: A21071 | IF (1:1000) |
| Antibody | Donkey anti-mouse Alexa Flour 647 | Thermo Fisher Scientific | Cat: A31571 | IF (1:1000) |
| Cell line (*M. musculus*) | B16F10 | ATCC | CRL-6475 | |
| Cell line (*M. musculus*) | Neuro-2A neuroblast | ATCC | CCL-131 | |
| Cell line (*M. musculus*) | NIH3T3 filbroblast | ATCC | CRL-1658 | |
| Chemical compound, drug | NHS-XX-Biotin | Merck KGaA | Cat: 203188 | |
| Chemical compound, drug | Oregon-Green-488 iodoacetamide | Invitrogen | Cat: O6010 | |
| Chemical compound, drug | Ni2+-NTA-agarose beads | Qiagen | Cat: 30230 | |
| Chemical compound, drug | tetramethylrhodamine (TAMRA)—5-maleimide | Invitrogen | Cat: T6027 | |
| Chemical compound, drug | methoxy-poly (ethylene glycol)-silane | Laysan Bio Inc | | |
| Chemical compound, drug | biotin-poly (ethylene glycol)-sil | Laysan Bio Inc | | |

*Continued on next page*

*Continued*

| Reagent type (species) or resource | Designation | Source or reference | Identifiers | Additional information |
|---|---|---|---|---|
| Chemical compound, drug | AquaMount | Thermo Fisher Scientific | Cat: 14-390-5 | |
| Chemical compound, drug | Alexa Flour 568-phalloidin | Thermo Fisher Scientific | Cat: A12380 | IF (1:1000) |
| Chemical compound, drug | Formaldehyde 37% | Sigma Aldrich | Cat: 252549 | |
| Commercial assay or kit | Lipofectamine RNAiMAX | Thermo Fisher Scientific | Cat: 137780–0775 | |
| Commercial assay or kit | Lipofectamine 3000 | Thermo Fisher Scientific | Cat: L2000-015 | |
| Commercial assay or kit | Pierce ECL Western Blotting Substrate detection kit | Thermo Fisher Scientific | Cat: 34580 | |
| Other | DMEM-Dulbecco's Modified Eagle Medium | Gibco BRL Life Technologies | Cat: 11995–073 | |
| Other | FBS-Fetal Bovine Serum | Sigma Aldrich | Cat: F9423 | |
| Other | 200 mM L-glutamine | Thermo Fisher Scientific | Cat: 25030–081 | |
| Peptide, recombinant protein | N-terminal HiLyte488 mTwf1 Tail | Anaspec | | |
| Peptide, recombinant protein | CARMIL CPI | WatsonBio Sciences | | |
| Peptide, recombinant protein | CARMIL CSI | WatsonBio Sciences | | |
| Peptide, recombinant protein | mTwf1 A305-D350 | WatsonBio Sciences | | |
| Peptide, recombinant protein | mTwf1 A305-D350, K325A | WatsonBio Sciences | | |
| Peptide, recombinant protein | PreScission protease | GE Healthcare | Cat: GE27-0843-01 | |
| Recombinant DNA reagent | chicken CPα1β1 | *Soeno et al., 1998* *Soeno et al., 1998* | | Plasmid |
| Recombinant DNA reagent | chicken SNAP-CPα1β1 | *Bombardier et al., 2015* *Bombardier et al., 2015* | | Plasmid |
| Recombinant DNA reagent | mouse CPα1β2 | *Kim et al., 2012* | | Plasmid |
| Recombinant DNA reagent | mouse CPα1Δ28 | *Kim et al., 2012* | | Plasmid |

*Continued on next page*

Continued

| Reagent type (species) or resource | Designation | Source or reference | Identifiers | Additional information |
|---|---|---|---|---|
| Recombinant DNA reagent | mouse CP α1β2 R15A/Y79A | *Edwards et al., 2015* *Edwards et al., 2015* | | Plasmid |
| Recombinant DNA reagent | human CARMIL1 CBR115 (964–1078) | *Kim et al., 2012* | | Plasmid |
| Recombinant DNA reagent | human V-1 | *Edwards et al., 2015* | | Plasmid |
| Recombinant DNA reagent | CARMIL1 | *Edwards et al., 2013* *Edwards et al., 2013* | | Plasmid |
| Recombinant DNA reagent | pGEX-6p-1-mTwf1 | *Hilton et al., 2018* *Hilton et al., 2018* | | Plasmid |
| Recombinant DNA reagent | pGEX-6p-1-mTwf2a | *Hilton et al., 2018* *Hilton et al., 2018* | | Plasmid |
| Recombinant DNA reagent | pGEX-6p-1-mTwf1-11 | This paper | | Plasmid |
| Recombinant DNA reagent | pEGFP-C1 | Clontech | | Plasmid |
| Recombinant DNA reagent | pCMV-M1 | Addgene | Cat: 23007 | Plasmid |
| Recombinant DNA reagent | pCMV-myc-mTwf1 | This paper | | Plasmid |
| Recombinant DNA reagent | pCMV-myc-mTwf1-11 | This paper | | Plasmid |
| Sequence-based reagent | siTwf1 | This paper | | siRNA; CGUUACCA UUUCUUUCUGUUU |
| Sequence-based reagent | siCP1 | This paper | | siRNA; CCUCAGCGA UCUGAUCGACUU |
| Sequence-based reagent | siCP2 | This paper | | siRNA; GCACGC UGAAUGAGAUCUA |
| Sequence-based reagent | control RNAi oligos (Stealth RNAi) | Invitrogen | Cat: 12935–200 | |
| Software, algorithm | Fiji/Image J | *Schindelin et al., 2012* | | |
| Software, algorithm | NIS Elements software - Version 4.30.02 | Nikon Instruments | | |
| Software, algorithm | GraphPad Prism 6.0 | GraphPad Software | | |
| Software, algorithm | Adobe Creative Cloud Illustrator | Adobe Systems | | |
| Strain, strain background (E. coli) | BL21 (DE3) pLysS | This paper | | |
| Strain, strain background (E. coli) | BL21 (DE3) pRIL | This paper | | |
| Strain, strain background (E. coli) | BL21 (DE3) pRARE | This paper | | |

## Plasmids

Plasmids used for expressing the following proteins were previously described: chicken CPα1β1 (*Soeno et al., 1998*), chicken SNAP- CPα1β1 (*Bombardier et al., 2015*), mouse CPα1β2 (*Kim et al., 2012*), mouse CPα1Δ28 (*Kim et al., 2012*), mouse CP α1β2 R15A/Y79A (*Edwards et al., 2015*), human CARMIL1 CBR115 (964–1078) (*Kim et al., 2012*), human V-1 (*Edwards et al., 2015*). The plasmid for over-expressing CARMIL1 in mammalian cells has been described (*Edwards et al., 2013*). To generate plasmids for expressing mouse Twinfilin isoforms as glutathione-S-transferase (GST)-fusions in *E. coli*, ORFs were PCR amplified from pHAT2-mTwf1 and pHAT2-mTwf2a kindly provided by Pekka Lappalainen (Univ. of Helsinki) (*Nevalainen et al., 2009*), and subcloned into the EcoRI and NotI sites of pGEX-6p-1, yielding pGEX-6p-1-mTwf1 and pGEX-6p-1-mTwf2a. pGEX-6p-1-mTwf1-11 (K332A, R333A) was generated by site-directed mutagenesis of pGEX-6p-1-mTwf1. For V-1 fluorescence experiments, we used a previously demonstrated strategy of removing two surface cysteine residues to allow direct labeling on the single remaining cysteine (*Fujiwara et al., 2014*); this was achieved by performing site-directed mutagenesis on wild type pGEX-GST-V-1 plasmid to introduce two mutations (C45S, C83S). To generate an RNAi-refractive construct of mTwf1 for expression in cultured cells, the ORF of mTwf1 was PCR amplified from pGEX-6p-1 and subcloned into the HindIII and SacI sites of pEGFP-C1 (Clontech, Mountain View, CA). Then, site-directed mutagenesis was used to introduce silent mutations at specific nucleotides of the ORF (703, 709, 711, 715), and the RNAi-refractive ORF was subcloned into the EcoRI and NotI sites of pCMV-M1, a gift from Linda Wordeman (*Stumpff et al., 2008*) (Addgene plasmid # 23007), yielding pCMV-myc-mTwf1. Site-directed mutagenesis was performed on pCMV-myc-mTwf1 to generate mutant pCMV-myc-mTwf1-11 (K332A, R333A). All constructs were verified by DNA sequencing.

## Protein expression and purification

Rabbit skeletal muscle actin (RMA) (*Spudich and Watt, 1971*), was purified from acetone powder generated from frozen ground hind leg muscle tissue of young rabbits (PelFreez, Rogers, AR). Lyophilized acetone powder stored at −80°C was mechanically sheared in a coffee grinder, resuspended in G-buffer (5 mM Tris-HCl pH 7.5, 0.5 mM DTT, 0.2 mM ATP, 0.1 mM CaCl$_2$), and then cleared by centrifugation for 20 min at 50,000 × g. Actin was polymerized by the addition of 2 mM MgCl$_2$ and 50 mM NaCl and incubated overnight at 4°C. F-actin was pelleted by centrifugation for 150 min at 361,000 × g, and the pellet solubilized by Dounce homogenization and dialyzed against G-buffer for 48 hr at 4°C. Monomeric actin was then precleared at 435,000 × g, and loaded onto a S200 (16/60) gel filtration column (GE healthcare, Marlborough, MA) equilibrated in G-Buffer. Peak fractions containing actin were stored at 4°C. For labeling actin with biotin (*Breitsprecher et al., 2012*) or Oregon Green (OG) (*Kuhn and Pollard, 2005*), the F-actin pellet described above was Dounced and dialyzed against G-buffer lacking DTT. Monomeric actin was then polymerized by adding an equal volume of 2X labeling buffer (50 mM Imidazole pH 7.5, 200 mM KCl, 0.3 mM ATP, 4 mM MgCl$_2$). After 5 min, the actin was mixed with a 5-fold molar excess of NHS-XX-Biotin (Merck KGaA, Darmstadt, Germany) or Oregon-Green-488 iodoacetamide (Invitrogen, Carlsbad, CA) resuspended in anhydrous DMF, and incubated in the dark for 15 hr at 4°C. Labeled F-actin was pelleted as above, and the pellet was rinsed briefly with G-buffer, then depolymerized by Dounce homogenization, and dialyzed against G-buffer for 48 hr at 4°C. Labeled, monomeric actin was purified further on an S200 (16/60) gel filtration column as above. Aliquots of biotin-conjugated actin were snap frozen in liquid nitrogen and stored at −80°C. OG-488-actin was dialyzed for 15 hr against G-buffer with 50% glycerol and stored at −20°C.

For bulk actin assembly assays, RMA was fluorescently labeled with pyrenyl-iodoacetamide on cysteine 374 (Pollard and Cooper, 1984; Graziano et al., 2013). An RMA pellet stored at 4°C (prepared as described above) was dialyzed against pyrene buffer (25 mM Tris-HCl, pH 7.5, 100 mM KCl, 0.02% NaN3, 0.3 mM ATP, and 2 mM MgSO4) for 3–4 hr and then diluted with pyrene buffer to 1 mg/ml (23.8 µM). A sevenfold molar excess of pyrenyl-iodoacetamide was added, the actin solution was incubated overnight at 4°C, and aggregates were cleared by low-speed centrifugation. The supernatant (containing F-actin) was centrifuged for 3 hr at 4°C at 45,000 rpm in a Ti70 rotor (Beckman Coulter, Indianapolis, IN) to pellet F-actin. The actin pellets were disrupted by Douncing, dialyzed against G-buffer for 1–2 d, and gel filtered on a 16/60 S200 column. Peak fractions were pooled, aliquoted, snap frozen, and stored at −80°C.

Mouse non-muscle CPα1β2 was purified as described (*Graziano et al., 2014*). Briefly, the expression vector (*Soeno et al., 1998*) was transformed into *E. coli* strain BL21 pLysS. Cells were grown in LB to log phase, then expression was induced for 3 hr at 37°C by addition of 0.4 mM isopropyl-β-D-thiogalactopyranoside (IPTG). Cells were collected by centrifugation, washed with 25 ml water, and resuspended in lysis buffer (20 mM Tris pH 8.0, 1 mM EDTA, 0.1% Triton X-100, protease inhibitors) and lysed by lysozyme treatment and sonication. The cell lysate was clarified by centrifugation at 12,500 x g for 30 min at 4°C. Supernatants were loaded onto a 1 ml Q-HiTrap column (GE Healthcare) and eluted with a 45 ml salt gradient (0–500 mM KCl) in 20 mM Tris, pH 8.0. Peak fractions were pooled, concentrated using a centrifugal filter (Centiprep, MWCO 10 kDa; Millipore) to 3 ml, and loaded onto a 26/60 Superdex 75 gel filtration column (GE Healthcare) equilibrated in 50 mM KCl, 20 mM Tris, pH 8.0. Peak fractions were pooled and loaded onto a 5 ml Mono Q column (GE Healthcare) and eluted with a 30 ml salt gradient (0–500 mM KCl) in 20 mM Tris, pH 8.0. Peak fractions were pooled, dialyzed overnight at 4°C into HEK buffer (20 mM HEPES, pH 7.4, 1 mM EDTA, 50 mM KCl), aliquoted, snap-frozen in liquid N2, and stored at −80°C.

SNAP-649-CP (CPα1β1) was purified and labeled as described (*Bombardier et al., 2015*). SNAP-CP was expressed E. coli strain BL21 pLysS. Cells were grown to log phase at 37°C, and then expression was induced for 8 hr at 37°C by addition of 0.4 mM isopropyl-β-D-thiogalactopyranoside (IPTG). Cells were collected by centrifugation, and resuspended in 20 mM Tris pH 8.0, 1 mM EDTA, 0.1% Triton X-100, protease inhibitors and lysed by lysozyme treatment and sonication. The cell lysate was centrifuged for 80 min at 60,000 rpm, 4°C in a Ti70 rotor (Beckman/Coulter, Fullerton, CA). The supernatant was rotated with 0.75 ml of Ni2+-NTA-agarose beads (Qiagen, Valencia, CA). SNAP-CP was fluorescently labelled using 9 μM (~4-fold excess) dye adduct for 2 hr at room temperature, yielding SNAP-649-CP. To remove free dye, beads were washed three times with 20 mM imidazole (pH 8.0), 1X PBS, 1 mM DTT, 200 mM NaCl. Labeled SNAP-649-CP was eluted with 0.5 ml of 300 mM imidazole pH 8.0, 50 mM Tris pH 8.0, 100 mM NaCl, 1 mM DTT, 5% glycerol, then purified by gel filtration on a Superose six column (GE Healthcare) equilibrated in 20 mM Hepes (pH 7.5), 1 mM EDTA, 150 mM KCl, 5% glycerol. Peak fractions were pooled, concentrated, aliquoted, snap-frozen in liquid N2, and stored at −80°C.

For stopped-flow kinetics, fluorescence anisotropy binding and HDX-MS experiments, His-tagged-α1 and β2 subunits of mouse CP (pRSFDuet-1, pBJ 2041) were co-expressed in *E. coli* BL21 (DE3) pRIL and purified as described (*Johnson et al., 2018*). For CP lacking the β tentacle, a premature stop codon was introduced, so that the C-terminal residue of the mouse β2 subunit was L243 instead of C272 (pBJ 1891).

Twinfilin polypeptides were expressed as GST-fusions in E. coli strain BL21 pRARE. Cells were grown to log phase at 37°C, and then expression was induced for 16 hr at 18°C by addition of 0.4 mM isopropyl-β-D-thiogalactopyranoside (IPTG). Cells were collected by centrifugation, washed with 25 ml water, and resuspended in 10 ml of PBS supplemented freshly with 0.5 mM dithiothreitol (DTT), 1 mM phenylmethylsulphonyl fluoride (PMSF), and a standard mixture of protease inhibitors. Cells were incubated with lysozyme (0.5 mg ml−1) on ice for 15 min and then sonicated. The cell lysate was clarified by centrifugation at 12,500 g for 20 min and incubated at 4°C (rotating) for at least 2 hr with 0.5 ml glutathione–agarose beads (Sigma-Aldrich; St. Louis, MO). Beads were washed three times in PBS supplemented with 1M NaCl and then washed two times in PBS. Twinfilin was cleaved from GST by incubation with PreScission Protease (GE Healthcare; Marlborough, MA) overnight at 4°C (rotating). Beads were pelleted, and the supernatant was concentrated to 0.3 ml, and then purified further by size-exclusion chromatography on a Superose12 column (GE Healthcare) equilibrated in HEK buffer (20 mM Hepes pH 7.5, 1 mM EDTA, 50 mM KCl, 0.5 mM DTT). Peak fractions were pooled, concentrated, aliquoted, snap-frozen in liquid N2, and stored at −80°C.

CARMIL CBR115 and V-1 were purified from *E. coli* as above for mTwf1 proteins, except the GST tag was removed from V-1 by digestion with thrombin instead of PreScission protease. To purify and label V-1 (generating TAMRA-V-1) for fluorescence experiments, BL21 *E. coli* expressing pGEX-GST-V-1 (C45S, C83S) was lysed in a Microfluidizer (Microfluidics Corp.; Westwood, MA). Fusion protein was isolated on Glutathione Superflow Agarose (Thermo Fisher Scientific; Waltham, MA). The GST tag was cleaved by digestion with bovine thrombin (MP Biomedicals; Santa Ana, CA) overnight at 4°C, then separated from V-1 on a Sephacryl S-200 HR 16/60 column (GE Healthcare) equilibrated in 25 mM HEPES pH 7.0, 1 mM TCEP, 100 mM KCl, 1 mM NaN3. Residual GST was removed by re-incubating peak fractions with Glutathione Superflow Agarose. Purified V-1 (C45S, C83S) was then

labeled with tetramethylrhodamine (TAMRA)−5-maleimide (Invitrogen) overnight at 4°C. Excess TAMRA was removed by dialysis against 20 mM 3-(N-morpholino)propanesulfonic acid (MOPS) pH 7.2, 1.0 mM TCEP, 100 mM KCl, 1 mM NaN3. TAMRA-V-1 was stored at −70°C

The mTwf1 tail peptides used for anisotropy were sourced as follows: N-terminal HiLyte488 labeled mTwf1 (H317-D350) was purchased from Anaspec (Fremont, CA); unlabeled CARMIL1 CPI (G969-A1005), CARMIL1 CSI (M1019-M1037), mTwf1 (A305-D350) and mTwf1 (A305-D350, K325A), as well as N-terminal TAMRA labeled mTwf1 (A305-D350), were purchased from WatsonBio Sciences (Houston, TX).

## Bulk pyrene F-actin assembly assays

Pyrene actin assembly assays were performed as previously described (Chesarone-Cataldo et al., 2011), with slight modifications for monitoring uncapping. Reactions containing 2 µM G-actin (5% pyrene labeled), 25 nM CapZ, and variable concentrations of mTwf1 were mixed to a volume of 52 µl followed by addition of 3 µl of initiation mix (40 mM $MgCl_2$, 10 mM ATP, 1 M KCl). Fluorescence was monitored at excitation and emission wavelengths of 365 and 407 nm, respectively, in a fluorescence spectrophotometer (Photon Technology International; Lawrenceville, NJ). Acquisition was paused at 400 s, and 5 µl of CARMIL CBR (final concentration 250 nM) was spiked into the reaction, mixed rapidly by pipetting, and measurement was resumed.

For pyrene actin elongation assays (as in *Figure 4A and B*), 5 µl of freshly mechanically sheared F-actin (10 µM) was added to a mixture of the indicated proteins or control buffers, and then immediately mixed with 0.5 µM monomeric actin (10% pyrene labeled) in 60 µl reactions and monitored in a plate reader (Infinite M200; Tecan, Männedorf, Switzerland) at excitation and emission wavelengths of 365 and 407 nm, respectively.

## Fluorescence anisotropy

The following anisotropy experiments were performed in HEK buffer (20 mM HEPES pH 7.5, 1 mM EDTA, 50 mM KCl, 0.5 mM DTT). Reactions were incubated at room temperature for 15 min, and anisotropy was determined by measuring polarized emission intensities at 525 nm when excited at 497 nm using a fluorescence spectrophotometer (Photon Technology International). To compare mTwf1-tail binding to wild type and mutant CP (*Figure 1B*), HiLyte-488-mTwf1 tail peptide (100 nM) was mixed with different concentrations of wild-type or mutant CP. To compare the abilities of full-length wild type mTwf1 and mutant mTwf1-11 polypeptides to compete with labeled mTwf1-tail for binding CP (*Figure 1C*), HiLyte-488-mTwf1 tail peptide (100 nM) was mixed with 1 µM CP and variable concentrations of full-length mTwf1 polypeptides.

The following anisotropy experiments were performed in the indicated buffer, incubated at room temperature for 2 min, and anisotropy was determined by measuring polarized emission intensities at 525 nm when excited at 497 nm for HiLyte-488, or at 582 nm when excited at 552 nm for TAMRA. To compare mTwf1 tail peptide binding to wild type CP and mutant CP(RY) (*Figure 2B*), HiLyte-488-mTwf1 tail peptide (60 nM) was mixed with different concentrations of CP or CP(RY) in HEK buffer containing 0.005% TWEEN 20. To compare the abilities of unlabeled wild type and mutant mTwf1 tail peptides to compete with labeled mTwf1 tail peptide for binding to CP (*Figure 2C*), TAMRA-mTwf1 tail peptide (A305-D350, 40 nM) was mixed with 1 µM CP and varying concentrations of the unlabeled tail peptides (mTwf1 A305-D350 or mTwf1 A305-D350, K325A) in 20 mM MOPS (pH 7.2), 1 mM TCEP, 100 mM KCl, 1 mM NaN₃, 0.005% TWEEN 20. To test the abilities of different fragments of CARMIL to compete with mTwf1 tail peptide for binding CP, HiLyte-488-mTwf1 tail peptide (60 nM) was mixed with 240 nM CP and different concentrations of mouse CARMIL1 CBR (964–1078), CPI (969–1005), or CSI (1019–1037) in HEK buffer containing 0.005% TWEEN20.

## Stopped-flow fluorescence

For kinetic dissociation experiments (as in *Figure 4D and E*), an SX.18MV stopped flow instrument with Pro-Data SX software V2.2.27 (Applied Photophysics Ltd., Leatherhead, UK) was used. 100 nM TAMRA-V-1 was preincubated with 2 µM CPα1β2. At time zero, TAMRA-V-1:CP complex was rapidly mixed via stopped-flow with an equal volume of a solution containing 5 µM unlabeled V-1, along with varied concentrations of mTwf1 or mTwf1-11. Experiments were performed at 25°C in HEK buffer containing 0.005% TWEEN20. Excitation occurred at 505 nm, with emission detected using a

570 + nm band-pass filter. All concentrations of mTwf were performed in replicates of 5–10, and traces were averaged. Apparent dissociation rates were determined by fitting the averaged data (5 ms. - 120 s.) to a single exponential model using Pro-Data Viewer software V4.2.27 (Applied Photophysics Ltd.).

## Total internal reflection fluorescence (TIRF) microscopy

For all experiments, 24 × 60 mm coverslips (Fisher Scientific; Pittsburg, PA) were cleaned by successive sonications as follows: 60 min in detergent, 20 min in 1 M KOH, 20 min in 1 M HCl min, and 60 min in ethanol. Coverslips were then washed extensively with ddH$_2$O and dried in an N$_2$-stream. A solution of 80% ethanol pH 2.0, 2 mg/ml methoxy-poly (ethylene glycol)-silane and 2 µg/ml biotin-poly (ethylene glycol)-silane (Laysan Bio Inc.; Arab, AL) was prepared and layered on the cleaned coverslips (200 µl per coverslip). The coverslips were incubated for 16 hr at 70°C. To assemble flow cells, PEG-coated coverslips were rinsed extensively with ddH$_2$O and dried in an N$_2$-stream, then attached to a prepared flow chamber (Ibidi; Martinsried, German) with double sided tape (2.5 cm x 2 mm x 120 µm) and five min epoxy resin. Flow cells were prepared immediately before use by sequential incubations as follows: 3 min in HEK-BSA (20 mM Hepes pH 7.5, 1 mM EDTA, 50 mM KCl, 1% BSA), 30 s in Streptavidin (0.1 mg/ml in PBS), a fast rinse in HEK-BSA, and then equilibration in 1X TIRF buffer, pH 7.5 (10 mM imidazole, 50 mM KCl, 1 mM MgCl$_2$, 1 mM EGTA, 0.2 mM ATP, 10 mM DTT, 15 mM glucose, 20 µg/ml catalase, 100 µg/ml glucose oxidase, and 0.5% methylcellulose (4000 cP)). To initiate reactions, actin monomers (10% OG-labeled, 0.5% biotinylated) were diluted to 1 µM in TIRF buffer, and immediately transferred to a flow chamber. After several minutes, once the actin filaments reached an appropriate length (approximately 10 µm), the reaction mixture was replaced by flow-in. For depolymerization experiments, the solution was replaced with TIRF buffer lacking actin monomers, with or without Twinfilin and/or CP polypeptides. For uncapping experiments, the solution was replaced with TIRF buffer lacking actin monomers, with 3 nM SNAP-649-CP (100% labeled), and filaments were allowed to be capped for 3 min. Subsequently, the solution was again replaced with TIRF buffer lacking actin monomers, with or without 50 nM CARMIL CBR and/or variable concentration of Twinfilin polypeptides. Time-lapse TIRF microscopy was performed using a Nikon-Ti200 inverted microscope equipped with a 150 mW Ar-Laser (Mellot Griot; Carlsbad, CA), a 60X TIRF-objective with a N.A. of 1.49 (Nikon Instruments Inc.; New York, NY), and an EMCCD camera (Andor Ixon; Belfast, Northern Ireland). During recordings, optimal focus was maintained using the perfect focus system (Nikon Instruments Inc). Images were captured every 5 s. The pixel size corresponded to 0.27 µm.

Filament depolymerization rates were determined by tracing filaments in ImageJ (http://rsbweb.nih.gov/ij) and measuring the change in length of individual filaments for 15–20 min after flow-in, or until filaments disappeared. Differences in fluorescence intensity along the length of the filament provided fiduciary marks that allowed us to distinguish barbed- and pointed-ends. Filament uncapping was measured by monitoring the as the amount of time that SNAP-649-CP puncta remained associated with the barbed end of a filament after the addition of CARMIL to the reaction (with or without Twinfilin) and expressing it as a fraction of filaments that remained capped at a given time point. All results shown are data from at least two independent TIRF experiments.

## Cell culture, transfection, and RNAi silencing

Mouse B16-F10 (CRL-6475), Neuro-2a (CCL-131), and NIH/3T3 (CRL-1658) cells obtained directly from ATCC (American Type Culture Collection; Manassas, VA), where their identities were authenticated by short tandem repeat DNA profiling and where they were tested for mycoplasma contamination. Cells were used for experiments within one year. All cells were grown in DMEM (Gibco BRL Life Technologies; Carlsbad, CA) supplemented with 10% fetal bovine serum (FBS; Sigma) and 200 mM L-glutamine (Thermo Fisher Scientific) at 37°C and 5% CO$_2$.

All cell culture experiments were carried out in 6-well dishes that were initially seeded with 100,000 cells. To knockdown Twinfilin-1 or Capping Protein cells were transfected 24 hr after seeding with 30 pmol siRNA oligo using Lipofectamine RNAiMAX (Thermo Fisher Scientific) according to the manufacturer's instructions. RNAi oligos directed against the mouse Twinfilin-1 coding region targeting (siTwf1) 5'- CGUUACCAUUUCUUUCUGUUU −3'; and against the Capping Protein β subunit coding region targeting (siCP1) 5'- CCUCAGCGAUCUGAUCGACUU-3', or (siCP2) 5'- GCACGC

UGAAUGAGAUCUA-3'. Cells were transfected in parallel with control RNAi oligos (Invitrogen). For over expression experiments cultured cells were transfected using Lipofectamine 3000 (Thermo Fisher Scientific) according to the manufacturer's instructions 24 hr after seeding. For CARMIL over expression experiments, 5 µG of DNA was transfected, and for Twinfilin over expression experiments 1 µG of DNA was transfected.

## Antibodies

The rabbit anti-Twinfilin was a generous gift from Pekka Lappalainen (Univ. Helsinki) and used a dilution of 1:1000 for western blot detection and 1:100 in cultured cells. A mouse anti-Capping Protein (Development Studies Hybridoma Bank; Iowa City, IA) was used at a dilution of 1:2000 for western blot detection and 1:50 in cultured cells. Mouse anti-Flag (F3165, Sigma) and rabbit anti-Myc (GTX29106, GeneTex; Irvine, CA) was used at 1:5000 for western blot detection and 1:500 in cultured cells. Mouse and Rabbit horseradish peroxidase conjugated secondary antibodies (GE Healthcare) were used at a dilution of 1:10,000 for western blot detection. Secondary antibodies for immunofluorescence (Alexa Fluor 488 or 647) and Alexa Fluor 568-phalloidin (ThermoFisher) were used at a dilution of 1:1000.

## Immunostaining cells

For cell-staining experiments, 48 hr post transfection, the cells were re-plated on 3 × 1×1 mm glass coverslip (VWR International) that had been acid washed and coated with Laminin (Invitrogen) and allowed to adhere for 3–6 hr. Cells were fixed for 15 min with 4% paraformaldehyde in PBS at room temperature and then permeabilized for 15 min in permeabilization solution (0.5% Triton X-100 and 0.3 M glycine in PBS) at room temperature. Slips were then blocked in 3% BSA dissolved in PBST (1X PBS and 0.1% TWEEN 20) for 1 hr at room temperature, then incubated in primary antibody (in PBST) for 12 hr at 4°C. Coverslips were then washed three times with 1X PBST and incubated with secondary antibodies (in PBST) for 1 hr at room temperature. Slips were washed three times with PBST and two times with PBS, and subsequently mounted on to slides with AquaMount (Thermo Fisher Scientific). Cells were imaged on a Nikon i-E upright confocal microscope equipped with a CSU-W1 spinning disk head (Yokogawa, Tokyo, Japan), 60x oil objective (NA 1.4; Nikon Instruments), and an Ixon 897 Ultra-CCD camera (Andor Technology) controlled by NIS-Elements software. Maximum intensity projections and raw fluorescence values were measured using Fiji.

## Western blotting

To measure protein levels in cells after silencing and rescue, cells were harvest 48 hr after initial oligo transfection and incubated for 10 min at 4°C in RIPA buffer (50 mM Tris, pH 7.5, 150 mM NaCl, 1% NP-40, 0.5% Na-deoxycholate, 0.1% SDS, 2 mM EDTA, 50 mM NaF). Samples were incubated on ice for 30 min, vortexed every 10 min, then precleared by centrifugation at 20,800 x $g$ for 15 min at 4°C, quantified by Bradford assay, and immunoblotted. Proteins were detected using a Pierce ECL Western Blotting Substrate detection kit (Thermo Fisher Scientific). Bands were quantified using Image-Lab (Biorad).

## Hydrogen deuterium exchange mass spectrometry (HDX-MS)

HDX-MS was performed as described (*Johnson et al., 2018*). CP and Twf1 samples were buffer-exchanged with 1X phosphate saline buffer (PBS), pH 7.4. HDX was initiated by diluting samples (25 µM, 2 µL) 10-fold with 1XPBS prepared in $D_2O$ buffer, or 1XPBS $H_2O$ buffer for samples measured for no-deuterium control. At different time intervals (10, 30, 60, 120, 360, 900, 3600, and 14400 s), the labeling reaction was quenched by rapidly decreasing the pH to 2.5 with 30 µL of quench buffer (3 M urea, 1% trifluoroacetic acid, $H_2O$) at 4°C. The protein mixture was immediately injected into a custom-built HDX sample-handling device that enabled digestion with a column containing immobilized pepsin (2 mm ×20 mm) at a flow rate of 100 µL/min in 0.1% formic acid. The resulting peptic peptides were captured on a ZORBAX Eclipse XDB C8 column (2.1 mm ×15 mm, Agilent) for desalting (3 min). The C8 column was then switched in-line with a Hypersil Gold C18 column (2.1 mm ×50 mm, Thermo Fisher), and a linear gradient (4–40% acetonitrile, 0.1% formic acid, 50 µL/min flow rate, over 5 min) was used to separate the peptides and direct them to an LTQ-FTICR mass spectrometer (Thermo Fisher) equipped with an electrospray ionization source. Valves, columns, and

tubing for protein digestion and peptide separation were immersed in an ice-water bath to minimize back-exchange.

To map the peptic peptides, the digest, in the absence of HDX, was submitted to accurate mass analysis by LC–MS/MS with the LTQ-FTICR, and the peptic peptides identified using Mascot (Matrix Science). For samples that underwent HDX, raw mass spectra and peptide sets were submitted to HDX Workbench (Pascal et al., 2012) for calculation and data visualization in a fully automated fashion. Peptides for each run were assessed based on relative representation and statistical validation as implemented within HDX Workbench. Appropriate approach to determine statistical significance between these data is by using Tukey's multiple comparison test. A representative time point was manually selected, replicate data points from multiple samples at this time point used to conduct a one-way analysis of variance (ANOVA) the divergence between the means of the experiments were assessed. In instances with large differences, Tukey method was used to determine statistical significance if the resulting $P$ value is less than 0.05. In the case where there was a comparison between two experiments, a $t$-test was used. Only the top six peptides from each MS scan were used in the final analysis. The extent of HDX at each time point was calculated by subtracting the centroid of the isotopic distribution of the nondeuterated peptide from that of the deuterated peptide. The relative deuterium uptake was plotted versus the labeling time to yield kinetic curves (%D vs time). Error bars represent the results of $t$-tests between samples are shown above each time point to illustrate statistical significance. For comparison between apo states and the complexes, differences in HDX for all time points were calculated. Absolute differences in perturbation values larger than 5% D were considered significant. HDX values at 15 min time point were mapped onto the protein three-dimensional (3D) structure for data visualization. Peptide digestions were optimized under HDX assay conditions, and the mass calculations included accommodation for back exchange with solvent.

## Acknowledgements

We are grateful to Julian Eskin, Sean Guo, Silvia Jansen, M Angeles Juanes, and Daisy Leung for helpful discussions and/or comments on the manuscript. We thank Ms. S Smith and Ms. M Torres for general support and coordination. In addition, we thank Steve DelSignore for help with analysis of cultured cells, Alex Kozlov and Timothy Lohman for instrument access, assistance, and advice on stopped-flow experiments, and Marlene Mekel for assistance and advice with biochemistry experiments. This work was supported by a grant from the NIH (R01 GM063691) to BLG, a grant from the NIH (R35 GM118171) to JAC., by Brandeis NSF MRSEC DMR-1420382, by grants from NIH (P01AI120943 and R01AI123926) and a grant from the Department of the Defense (Defense Threat Reduction Agency grant HDTRA1-16-1-0033; C Basler PI) to GKA. We thank Dr. Michael L Gross for the use of the mass spectrometry facilities supported by P41GM103422 and for facilitating the mass spectrometry studies. The content of the information does not necessarily reflect the position or the policy of the federal government, and no official endorsement should be inferred.

## Additional information

### Funding

| Funder | Grant reference number | Author |
| --- | --- | --- |
| National Institutes of Health | R01 GM063691 | Bruce L Goode |
| Defense Threat Reduction Agency | HDTRA1-16-1-0033 | Gaya K Amarasinghe |
| National Institutes of Health | R35 GM118171 | John A Cooper |
| National Science Foundation | MRSEC DMR-1420382 | Bruce L Goode |
| National Institutes of Health | P01 AI120943 | Gaya K Amarasinghe |
| National Institutes of Health | R01 AI123926 | Gaya K Amarasinghe |

The funders had no role in study design, data collection and interpretation, or the decision to submit the work for publication.

## Author contributions
Adam B Johnston, Conceptualization, Data curation, Formal analysis, Validation, Investigation, Writing—original draft; Denise M Hilton, Conceptualization, Data curation, Formal analysis, Validation, Investigation, Writing—original draft, Writing—review and editing; Patrick McConnell, Britney Johnson, Data curation, Formal analysis, Validation, Writing—review and editing; Meghan T Harris, Avital Simone, Data curation, Formal analysis; Gaya K Amarasinghe, Conceptualization, Formal analysis, Supervision, Funding acquisition, Investigation, Writing—original draft, Project administration, Writing—review and editing; John A Cooper, Conceptualization, Supervision, Funding acquisition, Validation, Writing—original draft, Project administration, Writing—review and editing; Bruce L Goode, Conceptualization, Supervision, Funding acquisition, Writing—original draft, Project administration, Writing—review and editing

## Author ORCIDs
Adam B Johnston (iD) http://orcid.org/0000-0002-1210-4929
Denise M Hilton (iD) http://orcid.org/0000-0003-1577-1855
John A Cooper (iD) http://orcid.org/0000-0002-0933-4571
Bruce L Goode (iD) https://orcid.org/0000-0002-6443-5893

## Decision letter and Author response
Decision letter https://doi.org/10.7554/eLife.41313.017
Author response https://doi.org/10.7554/eLife.41313.018

# Additional files

## Data availability
All datasets associated with this article are included in the manuscript and supporting files.

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
