## [Decision Letter]

Thank you for submitting your article "A novel mode of Capping Protein-regulation by Twinfilin" for consideration by *eLife*. Your article has been reviewed by three peer reviewers, and the evaluation has been overseen by a Reviewing Editor and Anna Akhmanova as the Senior and Reviewing Editor. The following individuals involved in review of your submission have agreed to reveal their identity: Alphee Michelot (Reviewer #1); Robert Robinson (Reviewer #3). A further reviewer remains anonymous.

The reviewers have discussed the reviews with one another and the Reviewing Editor has drafted this decision to help you prepare a revised submission.

Summary:

This is an interesting manuscript that reports a significant advance in our understanding of the biochemical and cellular functions of the actin-binding protein twinfilin. The authors report that twinfilin has a capping protein (CP) interacting (CPI) motif, similar to the CPI motif in the CARMIL protein, which binds to and inhibits capping protein. Through a rigorous series of biochemical experiments, they show that twinfilin binds competitively with CARMIL to capping protein. However, twinfilin and CARMIL actually have opposite effect on the capping protein activity, and twinfilin promotes capping activity by competition with CARMIL. Twinfilin also promotes capping by accelerating capping protein dissociation from the inhibitor V-1. In cells, twinfilin and capping protein have similar activities, as silencing their expression causes similar phenotypes. This study identifies twinfilin as the first known example of a pro-capping interaction partner of capping protein, and suggests a revised model of capping protein regulation in cells.

All reviewers have very positively commented on the quality of the data and the writing of the manuscript. Below is the list of comments that can mostly be addressed by textual changes and some additional clarifications.

1) The authors demonstrate clearly that the twinfilin/CP interaction is highly conserved from yeast to mammals, and involves the same type of residues. However, the V-1/CP and CARMIL/CP interactions are much less conserved, questioning whether the proposed function for twinfilin as a pro-capper is indeed its main function. For example, there appear to be no proteins inhibiting CP in yeast. Do the authors have any strong argument in favor of a conserved pro-capping function for twinfilin?

2) Figure 2A – please explain how was the alignment created. The insertion of spaces looks arbitrary. Statistically, it is meaningless to align one long twinfilin sequence against several CPI sequences. One acceptable solution would be to remove all the spaces in the alignment. A second solution would be to align diverse CPI and twinfilin sequences using rigorous methods.

3) The authors should be careful with their definition of filopodia in describing the data shown in Figures 6 and 7. The cells in which capping protein or twinfilin have been silenced look smaller in area than control cells, with spike-like protrusions. The smaller area suggests possible cell rounding. Thus, the spikes could be filopodia, which are dynamic structures (cycle between protrusion and retraction) or they could be retraction fibers that result from cell rounding (do not actively protrude/retract).

4) Is twinfilin mislocalized in cells ectopically expressing CARMIL? Please explain or add some data.

5) Figure 3: Is CP 100% labeled in this experiment or do the authors only analyze filaments originally capped by CP? The authors should provide images for the control condition.

6) Figure 1: the authors should also provide kymograph examples of the barbed end dissociation assays.

7) Figure 7D: – This is the take home message of the manuscript. It nicely combines the biological activity with schematic representations of structural knowledge of the components. Two clarifications need to be included:

A) The action of CARMIL on CP post "transition" state is shown to promote filament dissociation – probably based on the data from Figure 1D,E, which was carried out in a background of zero G-actin. In the cell, where there is plenty of G-actin, the action of ectopic CARMIL (Figure 7A) causes filopodia, consistent with filament elongation. Likely both activities occur, possibly dependent on the proximity to the membrane and pro-elongation factors. The elongation-promoting properties of CARMIL should be worked into the figure, otherwise the figure is misleading.

B) The structural representation of the "capped" filament should be checked through making a model to be included in the supplementary information. The "actin"/CP interaction is known from the dynactin complex cryoEM structure. The CPI/CP interactions are known from crystallography, which can be used to place a model of the twinfilin tail. The single twinfilin domain/actin is known from crystallography and the location of the side binding sites can be inferred from the cofilin/actin cryoEM structure. Putting these elements together will verify the cartoon model of the "capped" filament. Specifically to be checked: Is the KPK motif running in the correct direction and is the tail long enough to reach to an actin-bound twinfilin domain? And are the twinfilin domain and CP compatible when bound to the same actin?

---

## [Author Response]

[…] All reviewers have very positively commented on the quality of the data and the writing of the manuscript. Below is the list of comments that can mostly be addressed by textual changes and some additional clarifications.1) The authors demonstrate clearly that the twinfilin/CP interaction is highly conserved from yeast to mammals, and involves the same type of residues. However, the V-1/CP and CARMIL/CP interactions are much less conserved, questioning whether the proposed function for twinfilin as a pro-capper is indeed its main function. For example, there appear to be no proteins inhibiting CP in yeast. Do the authors have any strong argument in favor of a conserved pro-capping function for twinfilin?

Thank you, this is an important point, and we have added a comment about this in the Discussion. Two recent studies showed that the *S. cerevisiae* protein Aim21 has a CPI motif and regulates CP function at cortical actin patches (Farrell et al., 2017; Shin et al., 2018). Further, we performed BLAST searches for additional CPI motif-containing proteins in the yeast proteome. Our search yielded several candidates, and we have purified two of them. Both yeast CPI proteins negatively regulate yeast CP activity in vitro, and yeast Twinfilin competes with both for CP binding and antagonizes the negative regulatory effects of one. Thus, the CPI family appears to extend from yeast to mammals. Our new story on yeast Twinfilin and CPI proteins is still a work in progress and will require at least another year of experiments before it is ready to publish. We have edited the Discussion by citing the two papers on Aim21, and mentioning that the CPI motif is conserved in yeast Twinfilin (Figure 2A).

2) Figure 2A – please explain how was the alignment created. The insertion of spaces looks arbitrary. Statistically, it is meaningless to align one long twinfilin sequence against several CPI sequences. One acceptable solution would be to remove all the spaces in the alignment. A second solution would be to align diverse CPI and twinfilin sequences using rigorous methods.

We replaced Figure 2A with a new alignment generated using the MAFFT algorithm in the DNASTAR Lasergene Suite/MegAlign Pro application. This alignment shows the homology between 8 evolutionarily diverse Twinfilins and 8 known CPI proteins (the same 8 previously shown).

3) The authors should be careful with their definition of filopodia in describing the data shown in Figures 6 and 7. The cells in which capping protein or twinfilin have been silenced look smaller in area than control cells, with spike-like protrusions. The smaller area suggests possible cell rounding. Thus, the spikes could be filopodia, which are dynamic structures (cycle between protrusion and retraction) or they could be retraction fibers that result from cell rounding (do not actively protrude/retract).

We have edited the manuscript (and the figures) for this nomenclature, emphasizing that silencing CP or Twinfilin leads to cells with excessive “microspikes” instead of claiming they are filopodia. We agree that CP and Twinfilin silenced cells have reduced cell area compared to control cells. However, when we replate cells after silencing they spread out on the coverslip, so we don’t think that they are rounding up. Thus, our observations are consistent with the idea raised by the reviewers, which is that instead of putting out filopodial protrusions these silenced cells may be failing to put out robust lamellipodia. We mention this in the Results, and thank the reviewers for raising this point.

4) Is twinfilin mislocalized in cells ectopically expressing CARMIL? Please explain or add some data.

We examined Twinfilin localization in cells ectopically expressing CARMIL, and it does not change. However, this is not necessarily unexpected, given that ectopically expressed CARMIL is restricted to the membrane at the leading edge, and has been shown not to alter CP localization (Edwards et al., 2013). Further, Twinfilin (alone or together with Srv2/CAP) can bind to barbed ends of filaments (and depolymerize them), so even if there is less active CP at the leading edge, this would not necessarily alter Twinfilin localization.

5) Figure 3: Is CP 100% labeled in this experiment or do the authors only analyze filaments originally capped by CP? The authors should provide images for the control condition.

The CP is 100% labeled, which was stated in the Materials and methods. However, for added clarity we have also included this information in the figure legend and Results. In addition, we added a better description of how these experiments were performed in the Results, explaining that: (1) we identified capped filaments prior to flow in of CARMIL and/or Twinfilin, and (2) we then recorded the dwell times of labeled CP molecules on barbed ends. We also added the requested control images to Figure 3B.

6) Figure 1: the authors should also provide kymograph examples of the barbed end dissociation assays.

We agree that the paper stands alone better if we include examples of the raw data for the barbed end dissociation assays. To address this, we added Video 1 showing barbed end depolymerization of filaments for the four conditions in Figure 1F: (a) buffer alone, (b) mTwf1, (c) wildtype CP + mTwf1, and (d) mutant CPα∆28 + mTwf1. In the original submission, we did not include this video only because similar videos had been included in another paper we recently published describing the barbed end depolymerization effects of mouse Twinfilin (mTwf) isoforms (Hilton et al., 2018).

7) Figure 7D: This is the take home message of the manuscript. It nicely combines the biological activity with schematic representations of structural knowledge of the components. Two clarifications need to be included:A) The action of CARMIL on CP post "transition" state is shown to promote filament dissociation – probably based on the data from Figure 1D,E, which was carried out in a background of zero G-actin. In the cell, where there is plenty of G-actin, the action of ectopic CARMIL (Figure 7A) causes filopodia, consistent with filament elongation. Likely both activities occur, possibly dependent on the proximity to the membrane and pro-elongation factors. The elongation-promoting properties of CARMIL should be worked into the figure, otherwise the figure is misleading.

We agree that ectopic CARMIL promotes uncapping, creating free barbed ends of filaments, which could then grow (e.g., if a formin or Ena/VASP comes on to drive processive elongation) or disassemble (e.g., if Twinfilin comes on to processively drive depolymerization). To incorporate this idea of the ‘elongation-promoting’ effect of CARMIL in our model, we have added an arrow that starts at the transition state (where CARMIL is uncapping the barbed end) and points back to the original state of filament assembly.

B) The structural representation of the "capped" filament should be checked through making a model to be included in the supplementary information. The "actin"/CP interaction is known from the dynactin complex cryoEM structure. The CPI/CP interactions are known from crystallography, which can be used to place a model of the twinfilin tail. The single twinfilin domain/actin is known from crystallography and the location of the side binding sites can be inferred from the cofilin/actin cryoEM structure. Putting these elements together will verify the cartoon model of the "capped" filament. Specifically to be checked: Is the KPK motif running in the correct direction and is the tail long enough to reach to an actin-bound twinfilin domain? And are the twinfilin domain and CP compatible when bound to the same actin?

As requested, we have put these elements together in a structural model of the ternary complex formed by Twinfilin, CP, and the barbed end of an actin filament (Figure 7—figure supplement 1). In the configuration shown, the KPK motif runs in the correct orientation, and the tail is long enough to reach the C-terminal ADFH domain of Twinfilin bound to the ultimate actin subunit in the filament. Further, there are no clashes between CP and Twinfilin on actin in this configuration. When we docked Twinfilin onto the opposite strand of F-actin, there was no longer sufficient length between the tail and the C-terminal ADFH domain to allow simultaneous binding of the tail (to CP) and the C-terminal ADFH domain (to actin). We mention this point in the legend.